# Partitioning $CO_2$ net ecosystem exchange fluxes on the pedon scale in the Lena River Delta, Siberia

Tim Eckhardt[1,2], Christian Knoblauch[1,2], Lars Kutzbach[1,2], David Holl[1,2], Gillian Simpson[3], Evgeny Abakumov[4] and Eva-Maria Pfeiffer[1,2]

[1]Institute of Soil Science, Universität Hamburg, Allende-Platz 2 Hamburg, 20146, Germany
[2]Center for Earth System Research and Sustainability, Universität Hamburg, Allende-Platz 2, Hamburg, 20146, Germany
[3]School of GeoSciences, University of Edinburgh, West Mains Road, Edinburgh, EH9 3JN, Scotland, UK
[4]Department of Applied Ecology, Saint-Petersburg State University, 199178, 16-line 2, Vasilyevskiy Island, Russia

*Correspondence to:* Tim Eckhardt (tim.eckhardt@uni-hamburg.de)

**Abstract.** Arctic tundra ecosystems are currently facing amplified rates of climate warming. Since these ecosystems store significant amounts of soil organic carbon, which can be mineralized to carbon dioxide ($CO_2$) and methane ($CH_4$), rising temperatures may cause increasing greenhouse gas fluxes to the atmosphere. To understand how net ecosystem exchange (NEE) of $CO_2$ fluxes will respond to changing climatic and environmental conditions, it is necessary to understand the individual responses of the processes contributing to NEE. Therefore, this study aimed to partition NEE at the soil-plant-atmosphere interface in an arctic tundra ecosystem and to identify the main environmental drivers of these fluxes. NEE was partitioned into gross primary productivity (GPP) and ecosystem respiration ($R_{eco}$) and further into autotrophic ($R_A$) and heterotrophic respiration ($R_H$). The study examined $CO_2$ flux data collected during the growing season in 2015 using closed chamber measurements in a polygonal tundra landscape in the Lena River Delta, northeastern Siberia. To capture the influence of soil hydrology on $CO_2$ fluxes, measurements were conducted at a water-saturated polygon center and a well-drained polygon rim. These chamber-measured fluxes were used to model the NEE, GPP, $R_{eco}$, $R_H$, $R_A$ and net primary production (NPP) at the pedon scale (1 m – 10 m) and to determine cumulative growing season fluxes. Here, the response of *in situ* measured $R_A$ and $R_H$ fluxes from permafrost-affected soils of the polygonal tundra to hydrological conditions have been examined. Although changes in the water table depth at the polygon center sites did not affect $CO_2$ fluxes from $R_H$, rising water tables were linked to reduced $CO_2$ fluxes from $R_A$. Furthermore, this work found the polygonal tundra in the Lena River Delta to be a net sink for atmospheric $CO_2$ during the growing season. The NEE at the wet, depressed polygon center was more than twice than at the drier polygon rim. These differences between the two sites were caused by higher GPP fluxes, due to a higher vascular plant density and lower $R_{eco}$ fluxes due to oxygen limitation under water-saturated conditions at the polygon center in comparison to the rim. Hence, soil hydrological conditions were one of the key drivers for the different $CO_2$ fluxes across this highly heterogeneous tundra landscape.

## 1 Introduction

An estimated 1,000 Petagrams of organic carbon (OC) are stored in the upper 3 m of northern permafrost-affected soils (Hugelius et al., 2014). Given the large amount of OC stored in these soils, the response of the arctic carbon cycle to a changing climate is of global importance (McGuire et al., 2009). Over thousands of years, carbon has been sequestered in permafrost-affected soils and sediments due to cold conditions and poor drainage resulting in

water saturation and slow organic matter decomposition. Currently, arctic ecosystems are facing amplified

warming (AMAP, 2017; Taylor et al., 2013), which will lead to the longer and deeper thawing of permafrost-affected soils (Romanovsky et al., 2010). On the one hand, the microbial decomposition of newly available thawed permafrost organic matter releases carbon dioxide ($CO_2$) and methane ($CH_4$) (e.g. Knoblauch et al., 2018; Knoblauch et al., 2013; Zimov et al., 2006a; Schuur et al., 2009; Grosse et al., 2011). On the other hand, higher temperatures increase the assimilation of $CO_2$ by tundra vegetation due to a prolonged growing period and

increased nutrient availability in the deeper layers of thawed soils (e.g. Beermann et al., 2017; Elmendorf et al., 2012; Salmon et al., 2016; Parmentier et al., 2011).

With an area of 3 million $km^2$, more than half of the northern high latitude tundra ecosystems are situated in Russia (Walker et al., 2005). To date, just a few studies on $CO_2$ fluxes from the vast Russian arctic tundra ecosystems are available (e.g. Parmentier et al., 2011; Marushchak et al., 2013; Rößger et al., 2019; Kittler et al., 2016) especially

on the pedon scale (Kwon et al., 2016; Corradi et al., 2005; Heikkinen et al., 2004; Zamolodchikov et al., 2000). Since tundra soils are highly heterogeneous on the pedon scale in terms of temperature and moisture (Aalto et al., 2013), measurements on this scale are required to determine the response of individual $CO_2$ fluxes to these parameters. To cover this heterogeneity on the pedon scale chamber measurements are more appropriate than the eddy covariance (EC) method, which cover the next larger scale, even though a downscaling EC approach for $CO_2$

fluxes of an arctic ecosystem was recently presented (Rößger et al., 2019). An improved understanding of $CO_2$ dynamics in permafrost-affected soils is needed to improve estimates of future $CO_2$ balances of the highly heterogeneous arctic tundra regions. Without developments in our understanding of the response of $CO_2$ dynamics in permafrost-affected ecosystems on changing climatic conditions such as temperature and moisture, estimates of the carbon balance of the circum-arctic tundra and its future response to changing climatic conditions remain

biased.

The net ecosystem exchange (NEE) of $CO_2$ between the land surface and the atmosphere is composed of the $CO_2$ uptake by plants, termed gross primary productivity (GPP); and the release of $CO_2$ from soils and plants, ecosystem respiration ($R_{eco}$) (Chapin et al., 2006). The latter can be further split into autotrophic respiration by plants ($R_A$) and heterotrophic respiration ($R_H$) consisting of microbial soil organic matter (SOM) decomposition. In this study

the atmospheric sign convention is used, whereby a positive NEE defines a net release of $CO_2$ from the soil to the atmosphere and a negative sign defines a net uptake of $CO_2$ from the atmosphere.

In order to partition NEE into its underlying fluxes, measurements of GPP, $R_{eco}$, $R_A$ and $R_H$ are required. These individual process-based fluxes governing the $CO_2$ balance respond differently to changing climatic conditions such as temperature and moisture. For instance, it was shown that temperature changes in arctic soils could cause

a significant increase of the $CO_2$ uptake via GPP (Shaver et al., 1998; Oberbauer et al., 2007; Natali et al., 2012; Mauritz et al., 2017), which can be, beside others, attributed to shifts in vegetation composition (Elmendorf et al., 2012; Hudson et al., 2011) and increased nutrient availability (Johnson et al., 2000; Salmon et al., 2016; Beermann et al., 2015). Furthermore, the effect of drainage on GPP remain uncertain some studies found drainage of arctic soils to reduce GPP (Merbold et al., 2009; Chivers et al., 2009; Kwon et al., 2016), while other studies found

drainage to lead to a slight increase of GPP (Olivas et al., 2010; Kittler et al., 2016). The effect of increasing soil moisture on GPP differs between ecosystems (Mauritz et al., 2017; Olivas et al., 2010; c.f. Chivers et al., 2009). As respiratory processes are temperature-sensitive (Mahecha et al., 2010), the release of $CO_2$ by $R_{eco}$ increases in response to soil warming across arctic ecosystems (e.g. Hicks Pries et al., 2015; Oberbauer et al., 2007; Natali et al., 2015). An increase of $R_{eco}$ was also observed as a result of drainage of arctic soils and vice versa a decrease

with increasing water-saturation (Elberling et al., 2013; Mauritz et al., 2017; Chivers et al., 2009; e.g. Kwon et al., 2016; Olivas et al., 2010), due to the presence or absence of oxygen in drained soils (Hobbie et al., 2002). However, it was also shown that $R_{eco}$ fluxes could increase with increasing water-saturation due to higher soil temperatures in water-saturated soils (Zona et al., 2012), which highlights the interconnection of moisture and temperature in soils. In general, higher soil temperatures lead to a higher increase of $R_{eco}$ than of GPP, which causes a reduction

of the net $CO_2$ uptake (Parmentier et al., 2011; Oberbauer et al., 2007; Voigt et al., 2017; Mauritz et al., 2017). Also drainage of arctic soils causes a reduction of NEE (means less negative values) due to a higher increase of $R_{eco}$ than GPP (Merbold et al., 2009; Chivers et al., 2009; Kittler et al., 2016; Olivas et al., 2010), while the effect of increasing water-saturation of soils on NEE differs between arctic ecosystems (Chivers et al., 2009; Mauritz et al., 2017). Both soil temperature as well as moisture are predicted to change in the future due to increased

temperatures and precipitation in the pan-Arctic (Christensen et al., 2013). As $R_{eco}$ and GPP respond differently to temperature and moisture changes it is essential not only to focus on changes to NEE, but to gain a quantitative understanding of its components and their individual responses to environmental and climatic changes to improve model simulations of future $CO_2$ fluxes. Therefore, partitioning approaches of *in situ* measured $CO_2$ fluxes are required.

The release of $CO_2$ from soils by $R_{eco}$ is the largest efflux of carbon from terrestrial ecosystems to the atmosphere (Mahecha et al., 2010). Autotrophic respiration can be separated into aboveground plant respiration and belowground plant respiration (i.e. respiration of roots). Heterotrophic respiration is associated with the decomposition of SOM by heterotrophic soil organisms. To date, only a few estimates of $R_H$ fluxes from arctic tundra ecosystems over the growing season have been published (Nobrega and Grogan, 2008; Biasi et al., 2014;

Segal and Sullivan, 2014), with data lacking for ecosystems such as the polygonal tundra. Warming of the arctic soils will influence $R_H$ fluxes both directly and indirectly: rising soil temperatures will increase SOM decomposition, but will also cause permafrost thaw, exposing previously frozen SOM to microbial decomposition (Schuur et al., 2011; Dorrepaal et al., 2009). This decomposition could substantially reduce carbon storage in arctic tundra ecosystems, as gross ecosystem productivity has been found to be less temperature-sensitive than $R_{eco}$ in

these ecosystems (Grogan and Chapin, 2000; Dorrepaal et al., 2009). Warming could also reduce soil moisture (Suseela et al., 2012), and increase $R_A$ due to increasing aboveground biomass (Natali et al., 2012), which could lead to a lower contribution of $R_H$ to $R_{eco}$ (Hicks Pries et al., 2015; Chen et al., 2016). Furthermore, changes in soil moisture are known to affect microbial activity in soils directly with decreasing activity during times of high and low soil moisture and an optimum at moderate soil moisture conditions (Moyano et al., 2013). The increase of $R_A$

and $R_H$ fluxes due to warming might be compensated for by higher net primary production (Hicks Pries et al., 2013), but whether this compensation is valid for the entire growing season and across the, on pedon scale, highly heterogeneous arctic ecosystems remains uncertain. Furthermore, it remains uncertain how $R_A$ fluxes will respond to changing hydrological regimes as the impact of this parameter on $R_A$ fluxes have never been analyzed in tundra regions.

As changes in soil temperature and moisture can significantly alter the individual fluxes contributing to NEE, the current study aims to improve the current understanding of $CO_2$ flux dynamics in permafrost-affected ecosystems by (i) partitioning NEE into individual flux components: photosynthesis, ecosystem respiration as well as autotrophic and heterotrophic respiration at the pedon scale of the polygonal tundra and (ii) gaining insights into the response of these individual fluxes to different environmental parameters. Therefore, closed chamber

measurements were conducted at two sites in the polygonal tundra in northeastern Siberia over an almost complete

growing season. Finally, a $CO_2$ budget for a nearly complete vegetation period is determined for the two sites using data-calibrated flux models. These models were based on the time-sensitive bulk flux partitioning model by Runkle et al. (2013), which has been used in different arctic ecosystems (Helbig et al., 2017; Zona et al., 2014).

**2 Study site**

The investigation area is located on Samoylov Island in the southern central Lena River Delta, northeastern Siberia (72°22'N, 126°28'E – Figure 1). The Lena river forms the largest delta in the Arctic, which can be geomorphologically divided in river terraces of different ages and flood-plain levels (Schwamborn et al., 2002). The delta is located in the continuous permafrost zone with permafrost extending to depths of 300 to 500 m (Yershov, 1998) and relatively low mean annual soil temperatures of -7.8 °C at 1.7 m depth compared to other

arctic tundra sites (Boike et al., 2013). The study site has an arctic continental climate, characterized by low temperatures and low precipitation. The mean annual air temperature between 1998 and 2011 was -12.5 °C, and mean annual precipitation between 1981 and 2011 was 321mm (Pogoda i Klimat, 2016), while summer rainfall is 125 mm, ranging from 52 mm to 199 mm (Boike et al., 2013). Polar day lasts from 7 May until 8 August, and polar night lasts from 15 November to 28 January. Snowmelt usually starts in the first half of June, and the growing

season usually spans from around mid-June until mid-September.

The study site is covered by ice-wedge polygonal tundra on a Late-Holocene river terrace with elevations from 10 to 16 m above sea level on the eastern part of Samoylov Island. The development of polygonal structures has created depressed polygon centers surrounded by elevated polygon rims with elevation differences of about 0.5 m. Underlying permafrost prevents drainage in polygon centers resulting in water-saturated soils, anoxic soil

conditions at shallow depths, and significant amounts of soil organic carbon of around 33 kg m$^{-2}$ in the uppermost meter (Zubrzycki et al., 2013). In contrast, due to oxic conditions in the top-soil, the elevated polygon rim soils have accumulated less soil organic carbon of around 19 kg m$^{-2}$ (Zubrzycki et al., 2013). A land cover classification based on Landsat satellite imagery revealed that if excluding large thermokarst lakes the polygonal tundra on Samoylov Island consists of 65 % of dry tundra, 19 % of wet tundra and 16 % of small water bodies including

small ponds overgrown by vascular plants (Muster et al., 2012).

In this study, two different sites were investigated: i) a wet-depressed polygon center (wet tundra) and ii) its surrounding elevated polygon rim (dry tundra, 72°22,442 N; 126°29.828 E). These sites were located within the footprint area of an eddy covariance (EC) system where NEE of $CO_2$ was measured (Holl et al., 2019; Kutzbach et al., 2007b; Wille et al., 2008; Runkle et al., 2013). The maximum active layer depth (ALD) at the study site was

deeper at the polygon center (40 cm) than at the polygon rim (30 cm). The soils at the polygon centers were classified as *Histic* or *Reductaquic Cryosols* (WRB, 2014) with a water table close to the soil surface. Polygon rim soils were characterized by cryoturbation and therefore classified as *Turbic Glacic Cryosols* (WRB, 2014) with a water table just a few centimeters above the permafrost table. Total organic carbon (TOC) contents above 10% were found in the surface horizon above the cryoturbated horizons of the polygon rim, while high TOC contents

were found at the polygon center throughout the active layer (Zubrzycki et al., 2013). Vegetation on polygon rims is dominated by mosses (*Hylocomium splendens*, *Polytrichum spp.*, *Rhytidium rugosum*), some small vascular plants (*Dryas punctata* and *Astragalus frigidus*) as well as lichens (*Peltigera spp.*) and can be classified as non-tussock sedge, dwarf-shrub, moss tundra (Walker et al., 2005). The vegetation of the polygon centers were dominated by the hydrophilic sedge *Carex aquatilis*, which have in general much higher growth forms than at the

rim, and mosses (*Drepanocladus revolvens, Meesia triqueta, Scorpidium scorpioides*) and classified as sedge, moss, dwarf-shrub wetland Walker et al. (2005).

## 3 Methods

### 3.1 Meteorological data

Meteorological variables were recorded at 30 minute intervals at the nearby EC system and adjacent meteorological station 40 m southwest of the study site. Data collected were: air temperature (MP103A, ROTRONIC AG, Switzerland), air pressure (RPT410F, Druck Messtechnik GmbH, Germany) and photosynthetically active radiation (PAR; wavelength: 400 – 700 nanometers; QS2, Delta-T Devices Ltd., UK) as well as the incoming and reflected components of shortwave and longwave radiation, respectively (CNR 1, Kipp and Zonen, Netherlands). The radiative surface temperature ($T_{surf}$; in Kelvin (K)) was calculated as:

$$T_{surf} = \left(\frac{L\uparrow_B}{\varepsilon\,\sigma}\right)^{1/4} \tag{1}$$

where $L\uparrow_B$ is the upward infrared radiation (W m$^{-2}$), $\sigma$ is the Stefan-Boltzmann constant (W m$^{-2}$ K$^{-4}$), and the dimensionless emissivity $\varepsilon$ was assumed to be 0.98 after Wilber et al. (1999). Furthermore, soil temperature ($T_{soil}$) was measured at 2 cm soil depth in intervals of 30 minutes at an adjacent polygon rim and center.

### 3.2 Soil sampling and vegetation indices

Undisturbed soil samples were taken from the active layer at the polygon rim using steel rings (diameter 6 cm). At the water saturated polygon center, an undisturbed soil monolith was taken from the active layer using a spade and subsequently subsampled into four soil layers based on the degradation status of the organic matter. Coarse roots were removed, and soil samples were homogenized for analysis of soil water content (mass difference between wet and dried (105 °C) soil samples) and pH (CG820, Schott AG, Mainz, Germany). Total carbon and nitrogen

(N) contents (VarioMAX cube, Elementar Analysesysteme GmbH, Hanau, Germany), as well as total organic carbon (TOC) and total inorganic carbon contents (TIC, liquiTOC II, Elementar Analysesysteme GmbH, Hanau, Germany) were determined from dried (105 °C for more than 24 hours) and milled soil samples. To analyze vegetation indices, gridded quadrats of 10 cm x 10 cm were placed over the collars, and a visual identification of the plant species present as well as their abundance (% surface cover) was conducted in four grid squares.

### 3.3 Net ecosystem exchange and ecosystem respiration

A total of eight PVC frames (50 cm x 50 cm), four at each site, were installed in July 2014 in preparation for NEE and $R_{eco}$ flux measurements with closed chambers the following year. The frames were equipped with a U-shaped frame filled with water to avoid gas exchange between the chamber headspace and ambient air. The chamber (50 cm x 50 cm x 50 cm) used for NEE and $R_{eco}$ flux measurements was made of clear acrylic glass (Plexiglas

SunActive GS, Evonik Industries AG, Germany). The chamber was equipped with a fan for continuous mixing of headspace air (axial fan, 12V/DC, Conrad Electronic SE, Germany). Furthermore, a PAR sensor (SKP212, Skye Instruments Ltd., UK) and a temperature probe (107 Thermistor probe, Campbell Scientific Ltd., USA) were installed inside the chamber. Including the volume inside the chamber frames, the chamber enclosed a volume of 124 - 143 l. For $R_{eco}$ measurements, the chamber was covered with an opaque material. Boardwalks were installed

at both sites to avoid disturbance. The volumetric soil water content (VWC) was measured with a GS3 sensor

(Decagon Devices, Inc., USA) during each measurement directly beside the chamber frame at a depth of 5 cm. A diver (Schlumberger Ltd., USA) was installed at the polygon center to measure water table (WT) depth every 15 minutes. To prevent pressure-induced gas release during chamber closure (Christiansen et al., 2011), two holes (3 cm in diameter) at the top of the chamber were left open while placing the chamber on the frames and then closed for measurements. Soil temperatures between the surface and the frozen ground in 5 cm intervals and thaw depth were measured daily at both sites. For each chamber flux measurement, $CO_2$ concentrations in the chamber headspace were continuously measured with a gas analyzer (UGGA 30-p, Los Gatos Research, USA). The chamber headspace air was pumped in a closed loop via transparent polyurethane tubes (inner diameter 4 mm, each 10 m length) through the analyzer with a flowrate of 200 mL min$^{-1}$. The $CO_2$ concentration was logged (CR800series, Campbell Scientific Ltd., USA) together with PAR as well as soil and air temperature at a frequency of 1 Hz. Each chamber closure period was restricted to 120 sec to minimize warming inside the chamber relative to the ambient temperature.

Chamber measurements were conducted from 11 July until 22 September 2015, at least every third day between 6 am and 9 pm (local time), apart from the period 2-9 August and 17-24 August. Two consecutive measurements were performed at each frame: first, NEE (n = 679) was measured with the transparent chamber, followed by an $R_{eco}$ measurement (n = 679) with the dark chamber shortly after. The four frames of one site were measured consecutively before moving to the other site. GPP fluxes were calculated from the sum of the measured $R_{eco}$ and NEE fluxes.

### 3.4 Heterotrophic respiration

For $R_H$ measurements the root-trenching method was applied at both sites. It is challenging to separate belowground respiration fluxes into autotrophic and heterotrophic components because roots and microorganisms are closely linked within the rhizosphere (Hanson et al., 2000) . There are a wide range of methods for partitioning $R_{eco}$ (Subke et al., 2006; Kuzyakov, 2006), each with its associated advantages and disadvantages. Root trenching for example, despite some disturbance on the plant-soil interface, can give accurate estimates of the rates of $R_A$ and $R_H$ (Diaz-Pines et al., 2010) and produces similar results as a non-disturbing [14]C partitioning approach in an arctic tundra ecosystem (Biasi et al., 2014) and a partitioning approach based on [13]C (Chemidlin Prévost-Bouré et al., 2009). In this study, by inserting PVC frames below the main rooting zone at 20 cm deep into the soil, lateral roots were cut off. All living plant biomass including living moss tissue inside the frames was removed carefully in 2014. To prevent re-growth, the living plant biomass was removed periodically over the measurement period. This removal causes the die-off of roots, and in a period of days after the disturbance $R_H$ equals NEE. A total of eight frames, four at each site, were prepared for $R_H$ measurements. $R_H$ fluxes (n = 662) were measured during the same periods and with the same closure period as NEE and $R_{eco}$ measurements on unaltered plots.

To test if $R_H$ fluxes are biased due to the additional decomposition of residual roots, four additional PVC frames (two per site) were installed in 2015 following the sampling and preparation protocol of 2014. A total of 302 $R_H$ flux measurements were made on these newly installed plots. The difference between the mean $R_H$ fluxes of each single plot trenched in 2014 and those trenched in 2015 were analyzed using a Student's t-test.

$R_A$ fluxes at the unaltered sites were calculated by subtracting the mean $R_H$ fluxes measured at the trenched sites from the mean of the $R_{eco}$ fluxes at the unaltered sites of the same day. The calculated $R_A$ fluxes were summed with the calculated GPP fluxes to estimate the net primary productivity (NPP) fluxes.

### 3.5 Flux calculation

$CO_2$ fluxes ($\mu g\ CO_2\ m^{-2}\ s^{-1}$) were calculated using MATLAB® R2015a (The MathWorks Inc., Natick, MA, 2000) with a routine that uses different regression models to describe the change in the chamber headspace $CO_2$ concentration over time and conducts statistical analysis to aid model selection (Eckhardt and Kutzbach, 2016; Kutzbach et al., 2007a).

Due to possible perturbations while placing the chamber on the frame, the first 30 seconds of each 2-minute measurement period were discarded and the remaining 90 data points were used for flux calculations. The precision of the gas analyzer with 1 s signal filtering is < 0.3 ppm for $CO_2$ according to the manufacturer. The root mean square error (RMSE) did not exceed this value under typical performance of chamber measurements and the fitting of the linear and non-linear regression models. Higher RMSE values indicated failed model fitting or disturbed chamber measurements. Therefore, if RMSE exceeded 0.3 ppm, the concentration-over-time curve was re-inspected. Variation of PAR during chamber measurements due to shifts in cloud cover leads to irregular $CO_2$ concentration time series and perturbation of the calculated $CO_2$ fluxes (Schneider et al., 2012). These perturbed concentration time series show distinct autocorrelation of the residuals of the regression models and were filtered out by using a threshold for residual autocorrelation indicated by the Durbin-Watson test (Durbin and Watson, 1950). The flux curve was re-inspected if the RMSE exceeded 0.3 ppm or showed a distinct autocorrelation, to see if irregularities could be removed by adjusting the size of the flux calculation window. If irregularities could be removed by adjusting the size of the flux calculation window, the flux curve was re-calculated and if not, the measurement was discarded. Overall, about 3% (n = 47) of the $CO_2$ flux measurements (NEE, $R_{eco}$ and $R_H$ measurements) were discarded from the dataset because they did not meet the abovementioned quality criteria.

Studies have shown that $CO_2$ fluxes calculated with linear regression models can be seriously biased (Kutzbach et al., 2007a), while non-linear regression models significantly improve flux calculations (Pihlatie et al., 2013). However, we found that the temporal evolution of $CO_2$ concentration in the chamber was best modelled with a linear regression model, as determined by the *Akaike* Information Criterion corrected for small samples sizes ($AIC_c$) (Burnham and Anderson, 2004). This is in good agreement with other studies, which have shown that in some cases a linear regression model can produce a better $CO_2$ flux estimate for a non-linear concentration-over-time curve than a non-linear regression model (Koskinen et al., 2014; Görres et al., 2014).

### 3.6 Modelling $CO_2$ fluxes at the pedon scale

Different numerical models were fitted to the measured $R_{eco}$ and $R_H$ fluxes and to the calculated GPP fluxes to quantify seasonal GPP, $R_{eco}$, and $R_H$ fluxes. To calibrate the models, these were fitted to the GPP, $R_{eco}$, and $R_H$ fluxes. The resulting fitting parameters were used to reproduce the fluxes over the complete measurement period. Model calibration was done by applying a 15-day moving window over the measurement period moving in one day intervals. If less than eight chamber measurements were performed during these 15 days, the moving window was extended to 19 days. Subsequently, the modelled fluxes for each measurement plot were averaged for each site. $CO_2$ fluxes from each of the four measurement plots were used separately for model calibration and the summed fluxes were used to analyze differences between both sites using a student's t-test.

The empirical $Q_{10}$ model (van't Hoff, 1898) was fitted to the measured $R_{eco}$ and $R_H$ fluxes:

$$R_{eco,H} = R_{base} \times Q_{10}^{\frac{T_{a,surf,soil} - T_{ref}}{\gamma}} \tag{2}$$

where the (variable) fit parameter $R_{base}$ is the basal respiration at the reference temperature $T_{ref}$ (15 °C). The reference temperature and $\gamma$ (10 °C) were held constant according to Mahecha et al. (2010). $Q_{10}$ was a fit parameter describing the ecosystem sensitivity of respiration to a 10 °C change in temperature. For this study a fixed $Q_{10}$ value of 1.52 was used, which represents the seasonal mean value of the bulk partitioning model for the $CO_2$ fluxes in the EC footprint area (Runkle et al., 2013). Air temperature ($T_a$), surface temperature ($T_{surf}$), and soil temperature ($T_{soil}$) measured at a depth of 2 cm were tested as input variables.

The model calibration was done with MATLAB® R2015a (The MathWorks Inc., Natick, MA, 2000). The model parameters were estimated by nonlinear least-squares regression fitting (nlinfit function), and the uncertainty of the parameters were determined by calculating the 95% confidence intervals using the nlparci function. The selection of the best performing temperature as input variable for the $R_{eco}$ and $R_H$ model was based on comparing the $R^2_{adj}$ of the model runs with different temperatures as input variable. The selected input variable was chosen for all measurement plots of the same site

To estimate GPP, the measured $R_{eco}$ fluxes were subtracted from the measured NEE for each measurement plot. The rectangular hyperbola function was fitted to the calculated GPP fluxes as a function of PAR (in µmol m$^{-2}$ s$^{-1}$):

$$GPP = -\frac{P_{max} \times \alpha \times PAR}{P_{max} + \alpha \times PAR} \tag{3}$$

where the (variable) fit parameter $P_{max}$ was the maximum canopy photosynthetic potential (hypothetical GPP at infinite PAR). The values for the initial canopy quantum efficiency $\alpha$ (in µg m$^{-2}$ s$^{-1}$ / µmol m$^{-2}$ s$^{-1}$; initial slope of the GPP model at PAR = 0) were obtained from modelling the $CO_2$ fluxes with EC data (Holl et al., 2018). From the determined values when $\alpha$ was held variable, a function was formulated that accounts for the seasonality of $\alpha$ with specific values for each day of the growing season using the following function:

$$\alpha = b \times exp^{\left(-\frac{abs\left((x-c)^d\right)}{2 \times e^2}\right)} + f \tag{4}$$

where $b = 0.042$, $c = 209.5$, $d = 2$, $e = 25.51$, $f = 0.008$ and $x =$ day of year 2015. Afterwards, these values (variable on daily basis) were used for both sites to reproduce GPP fluxes from chamber measurements over the complete measurement period.

Although the transmissivity of the chamber material was high with > 90 % for wavelengths between 380 and 780 nano meter (Evonik, 2015), it caused a reduction in the amount of incoming radiation reaching, which could be further reduced based on the sun elevation. During the complete measurement period, the PAR values measured inside the chamber were on average 20 % lower than the PAR values measured outside the chamber (data not shown). Therefore, GPP modelling was conducted in two steps. First, the GPP model was calibrated using PAR values measured inside the chamber; and secondly, the reproduction of GPP fluxes over the growing season was carried out using PAR values measured outside the chamber. Without this two-step calibration the GPP fluxes would have been underestimated.

The NEE for both sites were calculated as the sum of the modelled GPP and $R_{eco}$ fluxes. The $R_A$ fluxes were calculated as the difference of the modelled $R_{eco}$ and $R_H$ fluxes. Furthermore, NPP was calculated from the sum of $R_A$ and GPP fluxes.

As both sites are within the footprint of an EC station, which determines $CO_2$ fluxes on a larger spatial scale (100 to 1000 meter), the resulting NEE from the modelling approach was compared with NEE of the same period obtained from EC measurements reported by Holl et al. (2019). For this upscaling, resulting NEE from the chamber model were weighted (*NEE$_{chamber}$*) based on the half-hourly relative contributions of the surface classes defined by Muster et al. (2012) to the EC footprint using the following equation:

$$NEE_{chamber} = NEE_C \times Cover_{wet} + NEE_R \times Cover_{dry} \qquad\qquad (5)$$

where $NEE_C$ and $NEE_R$ are the modelled half-hourly chamber NEE for polygon center and rim, respectively and $Cover_{wet}$ and $Cover_{dry}$ the relative contribution of the surface classes polygon center and rim, respectively to the EC footprint as given in Holl et al. (2019).

## 4 Results

### 4.1 Meteorological data, environmental conditions, and soil characteristics

The mean daily air temperature over the study period ranged from 23 °C to -2 °C (Figure 2a). The average air temperature in August 2015 (9 °C) was similar to the long-term mean air temperature for the period 1998-2011 (Boike et al., 2013). Compared to the long-term mean, it was about 1°C colder during July (9 °C), whereas September was around 2 °C warmer than the reference period (3 °C). The total precipitation from mid-July to end of September 2015 was 78 mm which is below the mean precipitation of 96 ± 48 mm between 2003 and 2010 (Boike et al., 2013).

From mid-July to the end of September 2015, soil temperatures at 2 cm depth at the polygon rim showed a higher diurnal variability than at the center. The highest soil temperatures were measured in mid-July and at the beginning of August. At the end of September, the temperatures became slightly negative (Figure 2b). At the polygon rim, the thaw depth increased from the beginning of the campaign in mid-July until mid-September to reach a maximum depth of 36 cm. Maximum thaw depth was reached at the polygon center much earlier in the season (mid-July) and remained relatively constant until mid-September. The water table depth at the polygon center was tightly coupled to rainfall. The VWC at 5 cm soil depth was on average 30% at the polygon rim, with highest values observed after rainfall events (Figure 2c). The daily averaged PAR values showed a strong seasonality with decreasing daily mean values towards the end of the season, although there was a period at the end of July with rather low daily averaged PAR values.

The total soil organic carbon content was lower at the polygon rim (2-12%) than at the polygon center (10-20%) and showed a decrease with depth, which was more pronounced at the polygon rim. The estimated SOC stocks within 30 cm depth were about 11 kg m$^{-2}$ and about 21 kg m$^{-2}$ at the polygon rim and center, respectively. The total inorganic carbon content was at both sites in each soil depth 0.2 %.

### 4.2 Chamber CO$_2$ fluxes

In general, the CO$_2$ uptake (NEE) at the polygon center was higher (means more negative values) than at the rim (Figure 3). In September both sites acted as small net CO$_2$ sources. The standard error of the flux calculation was around 3.5 and 2.3 µg CO$_2$ m$^{-2}$ s$^{-1}$ for the polygon center and rim, respectively, and decreased slightly towards the end of the season. In contrast to the NEE, the measured R$_{eco}$ fluxes were on average higher at the rim compared to the center. The highest ecosystem respiration fluxes of rim and center, were measured at beginning of August, when the air temperature exceeded 20 °C.

In general, the release of CO$_2$ by R$_H$ was higher at the polygon rim than at the center and showed no seasonality (Figure 3). An increase in R$_H$ fluxes after periodical re-clipping of the vegetation were not observed. Comparing R$_H$ fluxes from measurement plots that were trenched in 2014 with those trenched in 2015 revealed no significant differences (t-test, $p > 0.05$) between the years of root-trenching (data not shown).

Due to a period with rather low daily averaged PAR at the end of July, the uptake was partly lower as at the beginning of the measurement period at both sites. After reaching peak net $CO_2$ uptake at the beginning of August, the uptake decreased until the end of September. This seasonality was more pronounced at the polygon center than at the polygon rim. Interestingly, towards September the net $CO_2$ uptake at the polygon rim exhibited an increase

for a period of about one week, before it decreased again towards the end of September. $R_{eco}$ fluxes showed a similar, but less distinct seasonal pattern, and the peak of the highest $R_{eco}$ fluxes was in mid-August. In contrast, $R_H$ fluxes showed no seasonal trend at the polygon center, while at the polygon rim the $R_H$ fluxes were also highest when $R_{eco}$ and NEE reached their maxima.

As GPP, NPP and $R_A$ fluxes were calculated from the measured NEE, $R_{eco}$ and $R_H$ fluxes, these fluxes show similar

patterns of seasonality. The highest GPP and NPP fluxes were observed during the vegetation maximum, with a more pronounced seasonality at the polygon center compared to the rim. In general, $R_A$ fluxes were within the same range at both sites which is in contrast to the calculated GPP fluxes, which were almost twice as high at the polygon center than at the rim.

Interestingly, the $R_{eco}$ fluxes were linearly correlated with WT fluctuations from the beginning of July until the

end of August (Figure 4d). In contrast, neither a trend of higher $R_H$ fluxes during times of high WT, nor lower $R_H$ fluxes during times of low WT were observed. Instead, the $R_A$ fluxes showed a significant correlation ($R^2 = 0.71$; $p < 0.05$) with WT fluctuations.

**4.3 Modelled $CO_2$ fluxes**

The fitting parameter of the GPP model (Equation 3), $P_{max}$, showed strong spatial and temporal variability (Figure

5b). The α values (Equation 4) used for the GPP model showed a high temporal variability with a mean of $1.47 \pm 0.62$. This value increased sharply towards the peak vegetation period at the end of July and decreased thereafter until the end of the growing season. The $P_{max}$ values showed a strong temporal variability (high standard deviation) at the polygon center (mean: $250.7 \pm 101.9$ μg $CO_2$ m$^{-2}$ s$^{-1}$). Considerable differences in $P_{max}$ were also observed between the polygon rim and the center. The average $P_{max}$ at the polygon rim

($135.4 \pm 37.2$ μg $CO_2$ m$^{-2}$ s$^{-1}$) was substantially lower than at the polygon center ($250.7 \pm 101.9$ μg $CO_2$ m$^{-2}$ s$^{-1}$). As with the measured NEE, $P_{max}$ values displayed an increase at the polygon rim towards the end of September. The fitting parameter of the $R_{eco}$ and $R_H$ model (Equation 2), $R_{base}$, also showed strong spatial and temporal variability (Figure 5d). In general, $R_{base}$ was higher at the polygon rim. The averaged $R_{base}$ values for the $R_H$ model fit differed substantially between sites with $14.6 \pm 2.1$ μg $CO_2$ m$^{-2}$ s$^{-1}$ at the polygon center and $29.0 \pm 2.9$

μg $CO_2$ m$^{-2}$ s$^{-1}$ at the polygon rim.

Polygon center $R_{eco}$ fluxes were best modelled using surface temperature as explanatory variable ($R^2_{adj} = 0.70$); while for the polygon rim the soil temperature showed the best fitting ($R^2_{adj} = 0.46$). In contrast to the $R_{eco}$ fluxes, the polygon center $R_H$ fluxes were best modelled when the air temperature was used as explanatory variable ($R^2_{adj} = 0.55$). At the polygon rim, using the soil temperature as explanatory variable showed the best fitting ($R^2_{adj} = 0.45$) when modelling $R_H$ fluxes. Differences in the goodness of the fits for the $R_{eco}$ flux model were small. The

$R^2_{adj}$ of the GPP model was 0.82 for the polygon center and 0.45 for the polygon rim.

The modelled GPP, $R_{eco}$ and $R_H$ fluxes were used to calculate the NEE, $R_A$ and NPP fluxes. All fluxes showed similar seasonal patterns as fluxes from chamber measurements. The comparison between modelled and measured fluxes showed highly significant correlation ($R^2 = 0.39 - 0.88$, $p < 0.001$, Figure 6 and 7). However, the fluxes at

the polygon rim tended to be underestimated by the model if the respiration fluxes were high and the other fluxes

were low (close to zero or positive NEE). A similar trend was observed for the respiration fluxes from the polygon center. Furthermore, NEE, GPP and NPP fluxes seem to be generally underestimated by the flux models. However, this offset was to be expected due to the use of different PAR values for flux calculation (see section 3.6).

**4.4 Integrated fluxes**

Based on the modelled chamber $CO_2$ fluxes, time-integrated $CO_2$ fluxes were calculated for the period between mid-July and end of September 2015 (Table 1, Figure 8). The integrated GPP flux at the polygon center was significantly (t-test, $p < 0.01$) higher than at the polygon rim. In contrast, the integrated $R_H$ fluxes at the polygon rim were almost double those at the polygon center ($p < 0.001$). This trend was also observed for $R_{eco}$ fluxes, although here the difference was not as large as seen for $R_H$ fluxes and was not significant ($p > 0.05$). Furthermore,

the flux differences in $R_A$ between the sites were rather small. Much higher GPP fluxes in association with lower $R_H$ and similar $R_A$ fluxes led to an integrated NEE, which was more than twice as high at the polygon center (-68 $\pm$ 12 $\mu g$ $CO_2$ $m^{-1}$ $s^{-1}$) than at the rim (-26 $\pm$ 19 $\mu g$ $CO_2$ $m^{-1}$ $s^{-1}$) and led to an almost twice as high NPP at the center than at the rim. The upscaled NEE from modelled chamber data correlated highly significant ($R^2 = 0.88$, $p < 0.001$) with modelled NEE from EC data (Figure 9). However, the upscaled NEE from modelled chamber data tended to

underestimate the highest uptake and release by NEE in comparison to modelled NEE from EC data.

**5 Discussion**

This study presented NEE, GPP, NPP as well as $R_{eco}$, $R_H$ and $R_A$ fluxes obtained from direct measurements and modelling approaches for dry and wet sites of the polygonal tundra. The $R_H$ fluxes were higher at the polygon rim compared to the center due to drier soil conditions at the rim. $R_A$ fluxes from both sites were similar although the

vascular plant cover at the center was higher, probably due to water-saturated conditions at the center. In addition, the integrated $R_{eco}$ fluxes at the rim were higher than at the center, due to higher $R_H$ and similar $R_A$ fluxes at both sites. The mean GPP fluxes are much higher at the center compared to the rim due to differences in vegetation between the sites. Together with $R_A$ fluxes that are within the same range between the sites, the differences in GPP lead to an almost two times higher NPP at the center compared to the rim. In sum, both the water-saturated polygon

center and the non-saturated polygon rim acted as net sinks for atmospheric $CO_2$ for the period mid-July to end of September 2015. However, the $CO_2$ sink strength differed substantially between wet and dry tundra, which can be related to the different hydrological conditions and vegetation composition

**5.1 $CO_2$ fluxes from arctic tundra sites**

To the best of our knowledge, $CO_2$ fluxes from polygon rim and center sites were reported merely from Barrow,

Alaska (Table 2). The daily averaged net $CO_2$ uptake at the polygon center from this study is twice as high as reported from any other study concerning $CO_2$ fluxes from polygonal tundra. Instead, beside this study, just the study by Olivas et al. (2011) reported the polygonal tundra to be a net sink, while other studies (Oechel et al., 1995; Lara et al., 2012; Lara and Tweedie, 2014) reported the polygonal tundra to be a net source of $CO_2$ over the growing season. The GPP fluxes from the polygon center from this study exceed the GPP fluxes from Barrow

reported by Oechel et al. (1995) and Lara et al. (2012), but are distinctly lower than those reported by Olivas et al. (2011) and Lara and Tweedie (2014). In terms of respiration, the $R_{eco}$ fluxes from this study at both sites are lower compared than the reported $R_{eco}$ fluxes from the polygonal tundra at Barrow. However, the inter-annual variability

of reported $CO_2$ fluxes from Barrow is rather high, which also could be caused by different vegetation and soil composition between the sites at Barrow.

A comparison of the $CO_2$ fluxes from the wet and dry site from this study with other wet and dry sites of the arctic tundra revealed rather low photosynthesis and respiration rates from the polygonal tundra on Samoylov Island (Table 2). The $R_{eco}$ fluxes from this study on both sites are lowest compared to other sites and the GPP fluxes of the polygon rim from this study are at the lower end compared to other dry sites, while the GPP fluxes of the polygon center are in between the fluxes from other wet sites. Only one study from a *Carex* shrub site in Cherskii

reported higher NEE (Kwon et al., 2016) compared to the polygon center from this study. Both the moderate GPP and low $R_{eco}$ fluxes at the polygon center lead to rather high net $CO_2$ uptake compared to other arctic tundra sites.

**5.2 Factors controlling $CO_2$ fluxes**

The rather moderate GPP and low $R_{eco}$ fluxes of the polygonal tundra on Samoylov Island compared to other arctic sites might be due to differences in vegetation composition, organic matter contents, low nutrient availability as

well as low temperatures and radiation at the study site. The polygonal tundra on Samoylov Island is considered as an ecosystem with rather moderate GPP due to its low vascular plant cover with a maximum leaf coverage of 0.3 (Kutzbach et al., 2007b). Mosses, which have a high coverage (> 0.9), were dominant at both sites and have a much lower photosynthetic capacity than vascular plants (Brown et al., 1980). In general, photosynthesis of vascular plants as well as respiration fluxes are lowered due to the low nutrient availability in arctic tundra

ecosystems (Shaver et al., 1998). A low nutrient availability is typical for most tundra soils due to water saturated conditions and low soil temperatures (Johnson et al., 2000). These conditions cause low microbial decomposition rates (Hobbie et al., 2002), which in turn result in a low supply of bioavailable nutrients (Beermann et al., 2015). However, following Sanders et al. (2010) the nitrogen turnover rates of the soils found at the study site can be estimated as rather low compared to other arctic tundra sites. Additionally, the long-term average net radiation at

the study site (June to August, 1999-2011) was 85 W m$^{-2}$ (1999-2011), which is lower than values reported from other arctic tundra sites in Alaska and Greenland (Boike et al., 2013; e.g. Wendler and Eaton, 1990; Oechel et al., 2014; Soegaard et al., 2001; Lynch et al., 1999). These factors might explain the comparatively low $R_{eco}$ and moderate GPP fluxes at the polygon rim and center compared to other arctic tundra sites.

The differences observed in GPP between the polygon rim and center can be related to the vascular plant coverage.

The polygon center had a much higher abundance of sedges, while the rim was moss-dominated and the sparsely spread vascular plants had shorter and fewer leaves. Therefore, the photosynthetic capacity is higher at the polygon center than at the rim, resulting in the center having a higher GPP. Additionally, limited water availability due to the elevation of the polygon rim caused moisture run-off, with a drier or desiccated moss layer which may have contributed to a lower GPP (Olivas et al., 2011). On the other hand, Olivas et al. (2011) found GPP fluxes to be

higher at a polygon rim than at a polygon center in the Alaskan coastal plains. They related low GPP fluxes at the polygon center to submersion of the moss layer and vascular plants. At the polygon center of the current study, the WT was frequently below the soil surface so that submersion of erect vascular plants was not regularly observed, and most part of the moss layer itself was not submerged. This difference in GPP between the Alaskan study sites (Olivas et al., 2011) and those presented in this study reveals the important influence, beside the vegetation

composition, of water level and its fluctuations throughout the season on $CO_2$ fluxes.

Differences in respiration fluxes between the wet and dry sites can be related to different soil conditions. The cold and water-logged conditions, typical for the polygon centers, reduced decomposition of SOM due to oxygen

limitation, causing low microbial activity and therefore low $R_H$ (Hobbie et al., 2002; Walz et al., 2017). Furthermore, moisture run-off at the rim created drier conditions in the topsoil, which increased soil oxygen availability and subsequently enhanced $R_H$ and $R_{eco}$ (Oechel et al., 1998). In addition, the stronger diurnal amplitude of the soil temperature at the polygon rim compared to the center led to higher daily soil temperatures. Both the increased temperatures and oxygen supply at the polygon rim relative to the center enhance microbial decomposition causing higher $R_H$ fluxes to be observed at the polygon rim. As such, the low $CO_2$ uptake (NEE) at the rim are caused not only by low GPP, but also by higher $R_{eco}$ fluxes compared to the center. The higher NEE at the polygon center compared to the rim is mainly driven by substantially higher GPP, and lower $R_H$ fluxes, which are due to differences in vascular plant cover, temperature and hydrology. This finding is in good agreement with Nobrega and Grogan (2008) who compared a wet sedge, dry heath, and mesic birch site and found that the highest $CO_2$ uptake at the wet sedge site was due to limited $R_{eco}$ associated with the water-logged conditions.

Measurements of $CO_2$ fluxes at the polygon rim showed an increase of net $CO_2$ uptake throughout September, whereas at the polygon center the NEE appeared to continuously decrease (lower net uptake of $CO_2$). This increase in late season NEE at the polygon rim cannot be explained by rising PAR or temperature, but may be related to the photosynthetic activity of mosses. At the study site, Kutzbach et al. (2007b) considered September as period where moss photosynthesis dominates GPP. During this time of the growing season, mosses can still assimilate substantial amounts of $CO_2$ because they tend to reach light saturation at lower irradiance (Harley et al., 1989). The photosynthetic activity of mosses declines rapidly when they face desiccation, because they cannot actively control their tissue water content (Turetsky et al., 2012). Additionally, it has been shown that mosses face light stress during times of high PAR (Murray et al., 1993). This light stress causes delayed senescence and more late-season photosynthesis (Zona et al., 2011). On Samoylov, the photosynthetic activity on the moss-dominated polygon rim is expected to be low during warm and dry periods such as those seen at the beginning of September 2015, and during times of high PAR. In contrast, with continuous rainfall, dew formation and the lower PAR observed in mid-September, the mosses on the polygon rim are likely to have resumed their metabolic activity, which led to increasing NEE at the rim. These findings are in good agreement with Olivas et al. (2011), who reported the highest contribution of mosses to GPP at the beginning and end of the growing season.

**5.3 Partitioning respiration fluxes in arctic tundra ecosystems**

To date only a few studies have estimated $R_H$ fluxes from arctic tundra ecosystems over a growing season under *in situ* conditions (Nobrega and Grogan, 2008; Biasi et al., 2014). Surprisingly, the differences in $R_H$ flux estimates reported in the literature and those presented in this study were rather low. Differences in $R_H$ fluxes measured with the trenching method may result from differences in the time between trenching and start of the measurement. Nobrega and Grogan (2008) for example started their $R_H$ measurements one day after clipping, while measurements in this study and that of Biasi et al. (2014) started about one year after treatment. Therefore, although these studies employed a similar partitioning approach for seasonal estimates of $R_H$ fluxes, any comparison must be made with caution. The few $R_H$ flux estimates in the literature from other arctic tundra sites were higher than the $R_H$ values from the Lena River Delta (0.5 ± 0.1 and 0.3 ± 0.02 g C m$^{-2}$ d$^{-1}$ at polygon rim and center, respectively). Higher growing season $R_H$ fluxes than found in this study (0.8-1.8 g C m$^{-2}$ d$^{-1}$) have been measured at a mesic birch and dry heath site at Daring Lake in Canada (Nobrega and Grogan, 2008) and at a bare peat site (1.0 g C m$^{-2}$ d$^{-1}$) in the subarctic tundra at Seida, Russia (Biasi et al., 2014). Both sites contained substantially higher amounts of SOC in the organic-rich layer than the soil at the polygon rim and were well-aerated compared

to the soil at the polygon center, both of which likely caused a higher organic matter decomposition rate and could explain the higher $R_H$ fluxes than found at the polygonal tundra sites. Similar $R_H$ fluxes to those reported in our study were measured at a wet sedge site in Daring Lake (0.4 g C m$^{-2}$ d$^{-1}$) (Nobrega and Grogan, 2008), where soil and environmental conditions like WT, ALD, soil temperature, vegetation and SOC were similar to the Samoylov sites and at vegetated peat sites in Seida (0.4-0.6 g C m$^{-2}$ d$^{-1}$) (Biasi et al., 2014). Despite these differences, the average contributions of $R_H$ to $R_{eco}$ of 42% at the center and 60% at the rim are in good agreement with those observed at Seida (37 – 64%) and Daring Lake (44 – 64%). Similar contributions have also been determined from arctic tussock tundra sites, where $R_H$ makes up approximately 40% of growing season $R_{eco}$ (Segal and Sullivan, 2014; Nowinski et al., 2010) and from a moist acidic tussock tundra site (Hicks Pries et al., 2013). In contrast to these results, in a subarctic peatland Dorrepaal et al. (2009) report a substantially higher contribution of $R_H$ to $R_{eco}$ of about 70 %. The different contribution of $R_H$ to $R_{eco}$ at the polygon rim and center on Samoylov Island can be related to differences in vascular plant coverage and moisture conditions between these sites. The higher GPP at the center relative to the rim also caused higher rates of $R_A$, in turn lowering the contribution of $R_H$ to $R_{eco}$. Additionally, anoxic soil conditions due to standing water, which characterized the polygon center, reduced SOM decomposition rates. Furthermore, Moyano et al. (2013) and Nobrega and Grogan (2008) have shown that consistently moderate moisture conditions, as at the polygon rim, promote microbial activity and therefore enable higher $R_H$ rates than at the center.

At the polygon center, the WT significantly correlated with $R_{eco}$ and $R_A$ fluxes, but no correlation between $R_H$ fluxes and WT was found. In contrast to this, none of the determined respiration fluxes ($R_{eco}$, $R_H$, $R_A$) correlated with VWC at the polygon rim, which might be due to a rather low range of VWC (28 – 34 %). The $R_A$ fluxes may be negatively affected by high WT due to submersion of the moss layer and partwise vascular leaves as submersion can lead to plant stress, reducing productivity and nutrient turnover (Gebauer et al., 1995). However, if $R_A$ fluxes would be reduced due to low photosynthetic activity, we would expect a correlation between GPP and $R_A$ fluxes, as observed at the polygon rim ($R^2 = 0.48$, $p < 0.05$) but not at the center ($R^2 = 0.01$, $p > 0.05$). Instead, only half as much $CO_2$ is released by $R_A$ at the center compared to the rim at similar GPP fluxes, as the GPP : $R_A$ ratio indicates (10.5 vs. 5.1 for the polygon center and rim, respectively). It is likely that $R_A$ is reduced due to the water-saturated soils as shown previously for $R_{eco}$ fluxes in the Arctic (e.g. Christensen et al., 1998) maybe due to slow diffusion under water-saturated conditions (Frank et al., 1996). Furthermore, it might be possible that $R_H$ fluxes are not affected by water table fluctuations as the decomposition of SOM could take place in deeper layers. This finding is in contrast to a set of studies which attributed correlations between $R_{eco}$ fluxes and WT fluctuations solely to the impact of oxygen availability on $R_H$ fluxes (Juszczak et al., 2013; Chimner and Cooper, 2003; Dorrepaal et al., 2009), or observed an impact of moisture conditions on $R_H$ fluxes across multiple peatland ecosystems (Estop-Aragonés et al., 2018), while another study has shown no effect between water table fluctuations and $R_{eco}$ fluxes (Chivers et al., 2009). However, the partitioning approach used in this study showed that $R_H$ fluxes are not responding to water table fluctuations. Instead the $CO_2$ release by $R_A$ is correlated with water table fluctuations. These findings show the importance of hydrologic conditions for $R_{eco}$ fluxes and the need for partitioning approaches to understand the response of the individual $R_{eco}$ fluxes to changing hydrologic conditions.

To determine the impact of hydrological conditions and temperature on the $R_H$ and $R_A$ fluxes, it would be useful to perform both warming and wetting experiments *in situ*. So far, although a number of studies have determined the temperature response of NEE, GPP, and $R_{eco}$ fluxes in arctic ecosystems with warming experiments (e.g. Natali et al., 2011; Frey et al., 2008; Voigt et al., 2017), much less research has focused on the response of $R_A$ and $R_H$

fluxes to increasing temperature (Hicks Pries et al., 2015). Wetting experiments in arctic tundra ecosystems to

determine the individual response of $R_A$ and $R_H$ fluxes to changing hydrological conditions are also lacking. As climate change will likely lead to strong changes in the hydrological regimes of Siberian tundra regions (Zimov et al., 2006b; Merbold et al., 2009), the responses of respiration fluxes to altered hydrological conditions should be addressed in future studies.

**6 Conclusion**

The contributions of GPP, $R_{eco}$, $R_H$ and $R_A$ fluxes to NEE in a drained (rim) and water-saturated (center) site in the arctic polygonal tundra of northeast Siberia have been quantified in this study. Both investigated sites acted as $CO_2$ sinks during the measurement period mid-July to end of September 2015. The polygon center was a considerably stronger $CO_2$ sink than the polygon rim. The main drivers behind these differences in $CO_2$ fluxes at the pedon scale were the higher GPP at the polygon center as well as lower $R_H$ fluxes at the polygon center. The substantial

differences in NEE between the dry and wet tundra sites highlight the importance of pedon scale measurements for reliable estimates of $CO_2$ surface-atmosphere fluxes from arctic tundra sites and the important role of soil moisture conditions on $CO_2$ fluxes. Hereby, it was shown that $R_A$ fluxes respond water table changes, with a low release of $CO_2$ by $R_A$ fluxes during times of a high water table. Therefore, future studies on $CO_2$ fluxes from arctic tundra ecosystems should focus on the role of hydrological conditions as a driver of these fluxes.


*Data availability.* All data sets shown are available at https://doi.pangaea.de/10.1594/PANGAEA.898876 (last access: 5 March 2019; Eckhardt et al., *in review).*

*Author contributions.* TE, CK, LK and EMP designed the study. GS and TE performed the chamber measurements

and laboratory analysis. DH and TE performed the visualization of flux comparisons. TE wrote the manuscript with contributions from all authors.

*Acknowledgements.* We would like to thank the members of the joint Russian-German field campaigns LENA 2014 and LENA 2015, especially Mikhail N. Gregoriev (Permafrost Institute, Yakutsk, Russia), Waldemar

Schneider and Günter Stoof (Alfred Wegener Institute for Polar and Marine Research, Potsdam, Germany) and the crew of the Russian research station Samoylov for logistical as well as technical support. We are grateful to Josefine Walz and Mercedes Molina Gámez for valuable help with chamber measurements, and Norman Roessger for intensive support on model development (all Institute of Soil Science, Universität Hamburg). This work was supported by the German Ministry of Education and Research (CarboPerm-Project, BMBF Grant No. 03G0836A

and the KoPf-Project, BMBF Grant No. 03F0764A). German co-authors got additional support from the Cluster of Excellence CliSAP (EXC177) at University of Hamburg funded by the German Research Foundation (DFG). We are also grateful for the reviews of Albertus J. Dolman and two anonymous reviewer and the comments of the editor Lutz Merbold on a previous version of this paper.

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

**Table 1 – Means and range of the modelled fluxes in µg $CO_2$ m$^{-2}$ s$^{-1}$.**

|  |  | polygon center | polygon rim |
|---|---|---|---|
|  |  | *in µg $CO_2$ $m^{-2}$ $s^{-1}$* | *in µg $CO_2$ $m^{-2}$ $s^{-1}$* |
| **NEE** | mean | -68 ± 12 | -26 ± 19 |
|  | range | -288 ± 53 to 54 ± 2 | -117 ± 60 to 49 ± 10 |
| **GPP** | mean | -98 ± 10 | -61 ± 17 |
|  | range | up to -342 ± 53 | up to -163 ± 57 |
| **R$_{eco}$** | mean | 29 ± 11 | 35 ± 9 |
|  | range | 12 ± 3 to 69 ± 7 | 21 ± 3 to 77 ± 14 |
| **R$_H$** | mean | 11 ± 3 | 21 ± 5 |
|  | range | 6 ± 1 to 27 ± 2 | 14 ± 4 to 46 ± 13 |
| **R$_A$** | mean | 19 ± 11 | 14 ± 5 |
|  | range | 1 ± 3 to 55 ± 4 | 5 ± 5 to 32 ± 19 |
| **NPP** | mean | -85 ± 12 | -49 ± 20 |
|  | range | up to -300 ± 53 | up to -142 ± 57 |


**Table 2 - Comparison of daily averaged $CO_2$ fluxes from different polygonal tundra sites, which are similar in vegetation and soil composition to our study site. All listed fluxes were measured with the closed chamber technique.**

| Location | Tundra type | Period | NEE (g C m$^{-2}$ d$^{-1}$) | GPP (g C m$^{-2}$ d$^{-1}$) | $R_{eco}$ (g C m$^{-2}$ d$^{-1}$) | Ref |
|---|---|---|---|---|---|---|
| *Lena River Delta, RU (72°N,127°E)* | *pol. rim* | *Jul-Sep 2015* | *-0.6 ± 0.4* | *-1.4 ± 0.4* | *0.8 ± 0.2* | *a* |
| | *pol. center* | | *-1.6 ± 0.3* | *-2.3 ± 0.2* | *0.7 ± 0.1* | |
| Barrow, US (71°N, 157°W) | pol. rim | Jun-Aug 2005 | -0.1 ± 0.5 | -3.7 ± 0.2 | 3.6 ± 0.3 | b |
| | pol. center | | -0.2 ± 0.2 | -3.1 ± 0.1 | 2.9 ± 0.1 | |
| | pol. rim | Jun-Aug 2006 | -0.7 ± 0.2 | -3.1 ± 0.3 | 2.4 ± 0.2 | |
| | pol. center | | -0.8 ± 0.2 | -2.3 ± 0.2 | 1.5 ± 0.2 | |
| Barrow, US (71°N, 157°W) | pol. center | Jun-Aug 1992 | 0.04 ± 0.05 | -0.8 ± 0.1 | 0.8 ± 0.1 | c |
| Barrow, US (71°N, 157°W) | pol. center | Jul-Aug 2008 | 0.1 ± 0.8 | -3.9 ± 1.8 | 3.9 ± 1.8 | d |
| Barrow, US (71°N, 157°W) | pol. center | Jul-Aug 2010 | 0.5 ± 0.8 | -1.7 ± 0.8 | 2.1 ± 1.2 | e |
| Daring Lake, CA (65°N, 111°W) | dry heath | Jun-Sep 2004 | -0.01 ± 0.1 | -1.7 ± 0.3 | 1.8 ± 0.2 | f |
| | wet sedge | | -0.9 ± 0.1 | -1.7 ± 0.1 | 0.8 ± 0.1 | |
| Cherskii, RU (68°N, 161°E) | carex shrub | Jul-Aug 2013 | -0.5 ± 0.1 | -2.5 ± 0.1 | 2.0 ± 0.1 | g |
| | | Jul-Aug 2014 | -2.2 ± 0.2 | -6.2 ± 0.1 | 4.0 ± 0.2 | |
| Vorkuta, RU (67°N, 63°E) | sedge bog | Jun-Aug 1996 | -1.0 ± 0.2 | -3.2 ± 0.4 | 2.2 ± 0.3 | h |
| Vorkuta, RU (67°N, 63°E) | wet tundra | Jun-Sep 2001 | -1.1* | -1.9* | 0.9* | i |
| | dry tundra | Jun-Sep 2001 | 1.2* | -1.9* | 3.2* | |
| Prudhoe Bay, US (70°N, 149°W) | wet tundra | Jun-Aug 1994 | -0.6 ± 0.4 | -5.2 ± 0.6 | 4.6 ± 0.3 | j |
| Lena River Delta, RU (72°N, 127°E) | dry tundra | Jun-Sep 2014 | -0.9 ± 3.0 | -3.6 ± 3.4 | 2.7 ± 0.9 | k |
| | | Jun-Sep 2015 | -0.7 ± 2.6 | -2.7 ± 3.2 | 1.9 ± 1.0 | |
| | wet tundra | Jun-Sep 2014 | -0.4 ± 1.9 | -2.3 ± 2.3 | 1.9 ± 0.7 | |
| | | Jun-Sep 2015 | -0.7 ± 2.4 | -2.9 ± 2.7 | 2.2 ± 0.7 | |

a: this study; b: Olivas et al. (2011); c: Oechel et al. (1995); d: Lara and Tweedie (2014); e: Lara et al. (2012); f: Nobrega and Grogan (2008); g: Kwon et al. (2016); h: Zamolodchikov et al. (2000); i: Heikkinen et al. (2004), *: standard deviation estimated; j: Vourlitis et al. (2000); k: (Rößger et al., 2019)

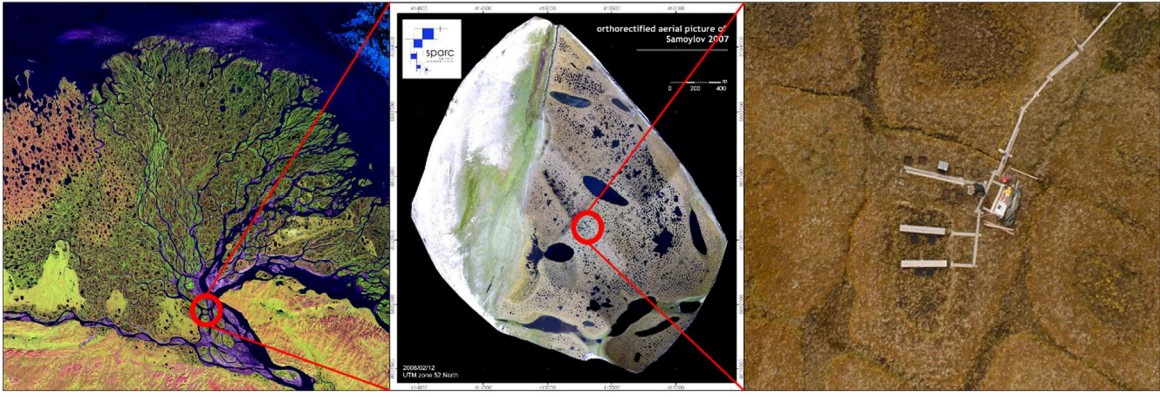

**Figure 1 - The study site on Samoylov Island, Lena River Delta in Northeastern Siberia (72°22'N, 126°28'E). (Satellite images – left: NASA (2002); middle: Boike et al. (2012); right: Boike et al. (2015)**

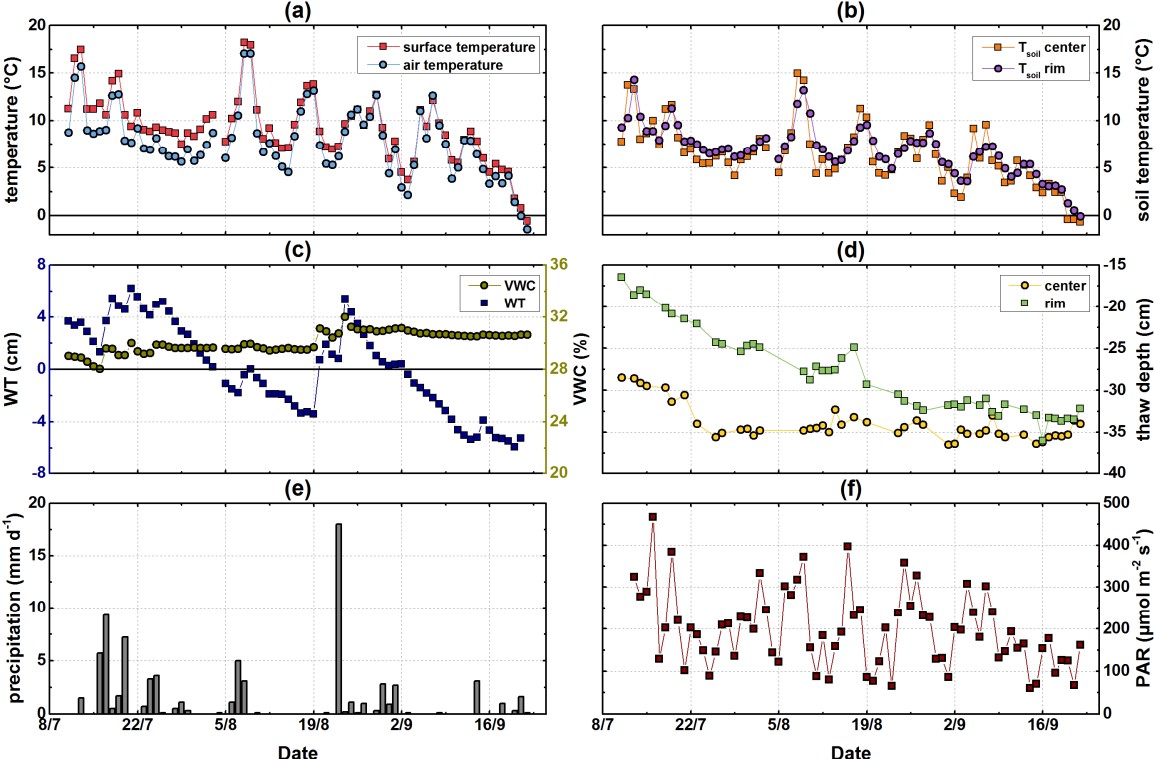

**Figure 2 - Meteorological conditions from mid July to end September. Panel (a) Half-hourly air temperature measured at 2 m height at the eddy covariance tower and surface temperature; (b) soil temperatures measured at 2 cm depth at polygon rim and center; (c) water table relative to the soil surface measured at the polygon center and volumetric water content measured at the polygon rim; (d) daily measured thaw depth at the polygon rim and center; (e) Daily precipitation measured at the eddy covariance station; (f) photosynthetically active radiation (PAR) measured half-hourly at the eddy covariance tower.**


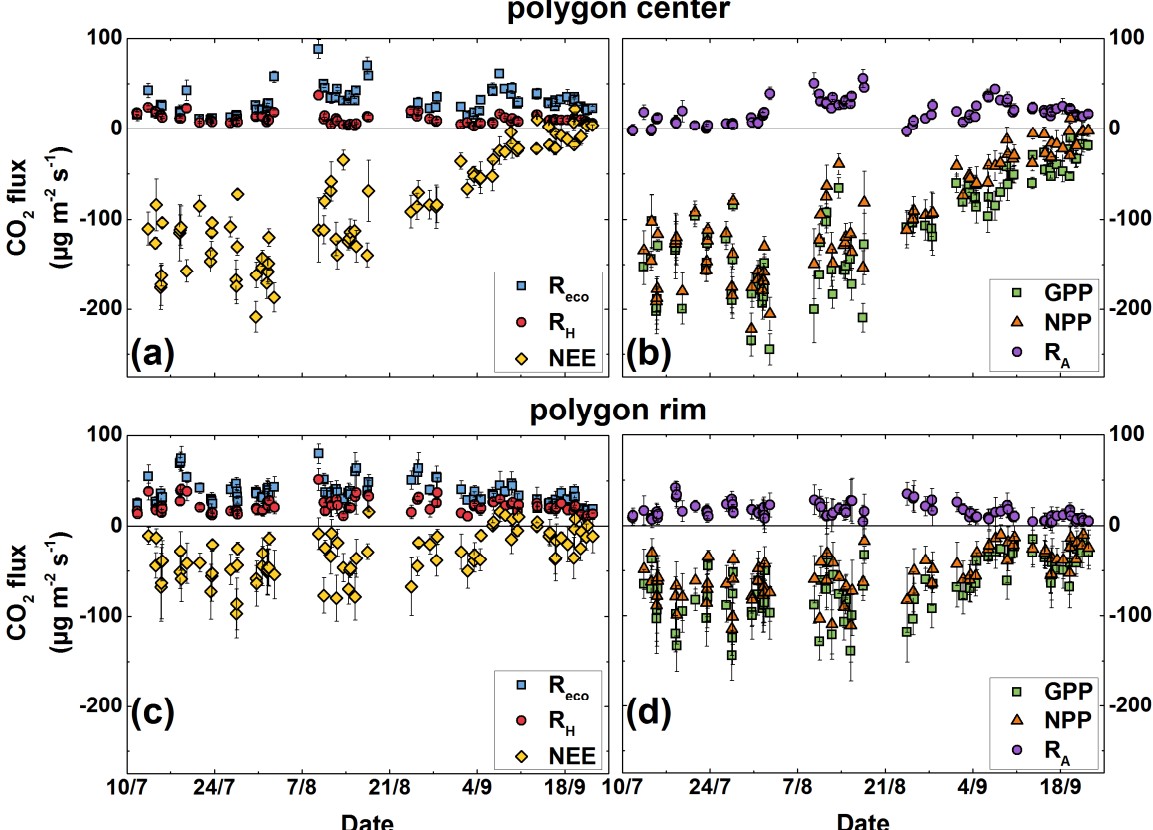

**Figure 3** - Chamber measured NEE, $R_{eco}$ and $R_H$ fluxes as well as calculated GPP, NPP and $R_A$ fluxes. The error bars denote the standard deviation of the four replicate measurements at each site. Panel (a) fluxes of NEE (n = 83), $R_{eco}$ (n = 85) and $R_H$ (n = 85) at the polygon center; (b) calculated fluxes of GPP (n = 83), NPP (n = 83) and $R_A$ (n = 85) at the polygon center; panel (c) measured fluxes of NEE (n = 83), $R_{eco}$ (n = 85) and $R_H$ (n = 85) at the polygon rim; (d) calculated fluxes of GPP (n = 83), NPP (n = 83) and $R_A$ (n = 85) at the polygon rim.

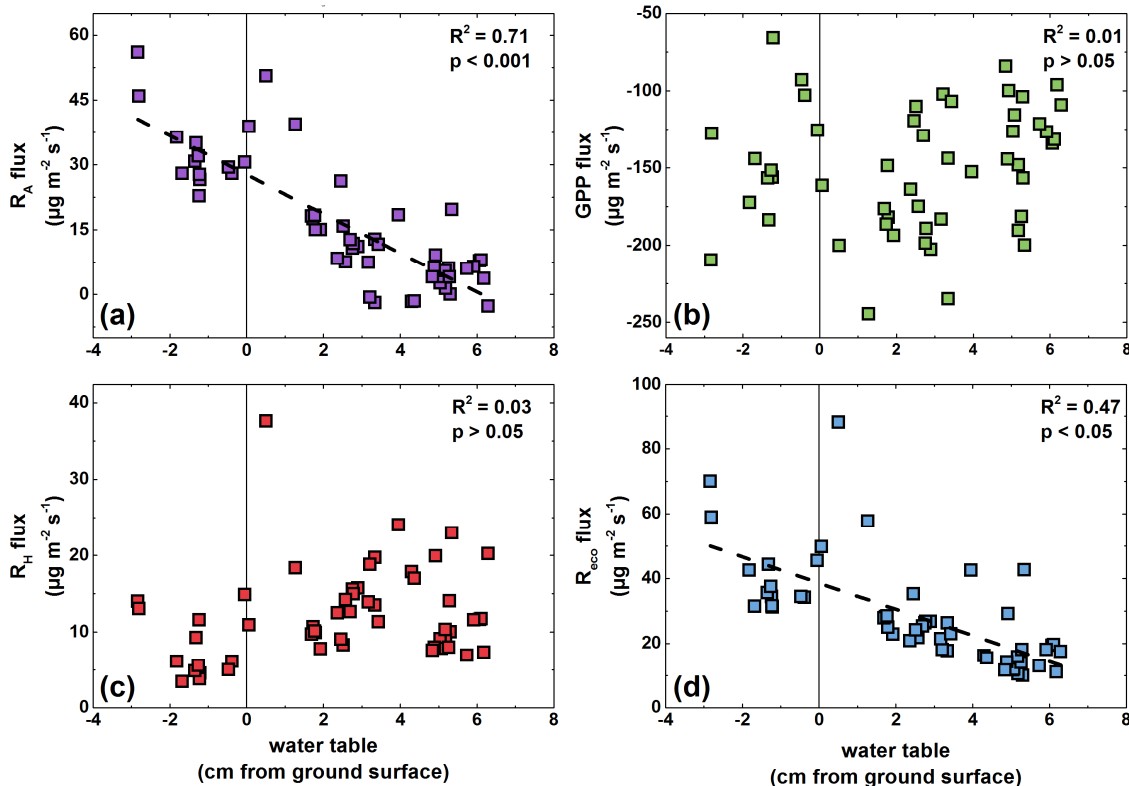


**Figure 4 - Relationships between water table fluctuations and (a) $R_{eco}$ fluxes, (b) $R_H$ fluxes, (c) $R_A$ fluxes and (d) GPP fluxes during the period July-August at the polygon center. Negative values on the x-axis indicate a water table below the soil surface.**

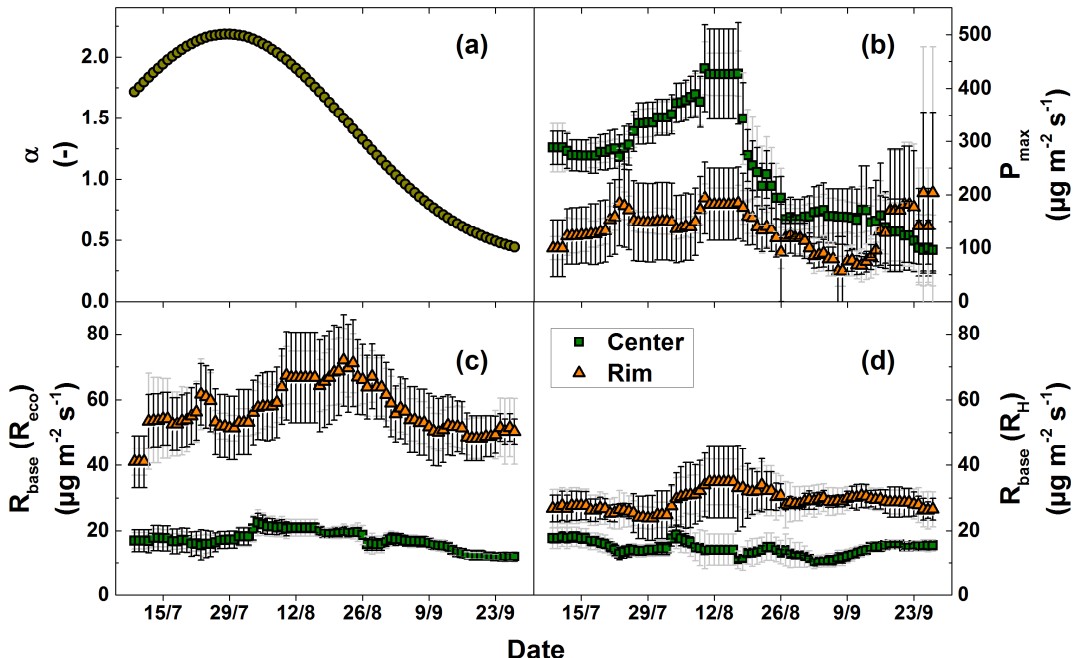


**Figure 5 - Fitting parameters of the $CO_2$ flux models. The values are given with the standard deviation of the model results from the single measurement plots (light grey error bars) and the confidence intervals (95%) of the fitting parameters (dark grey error bars).**

# polygon center

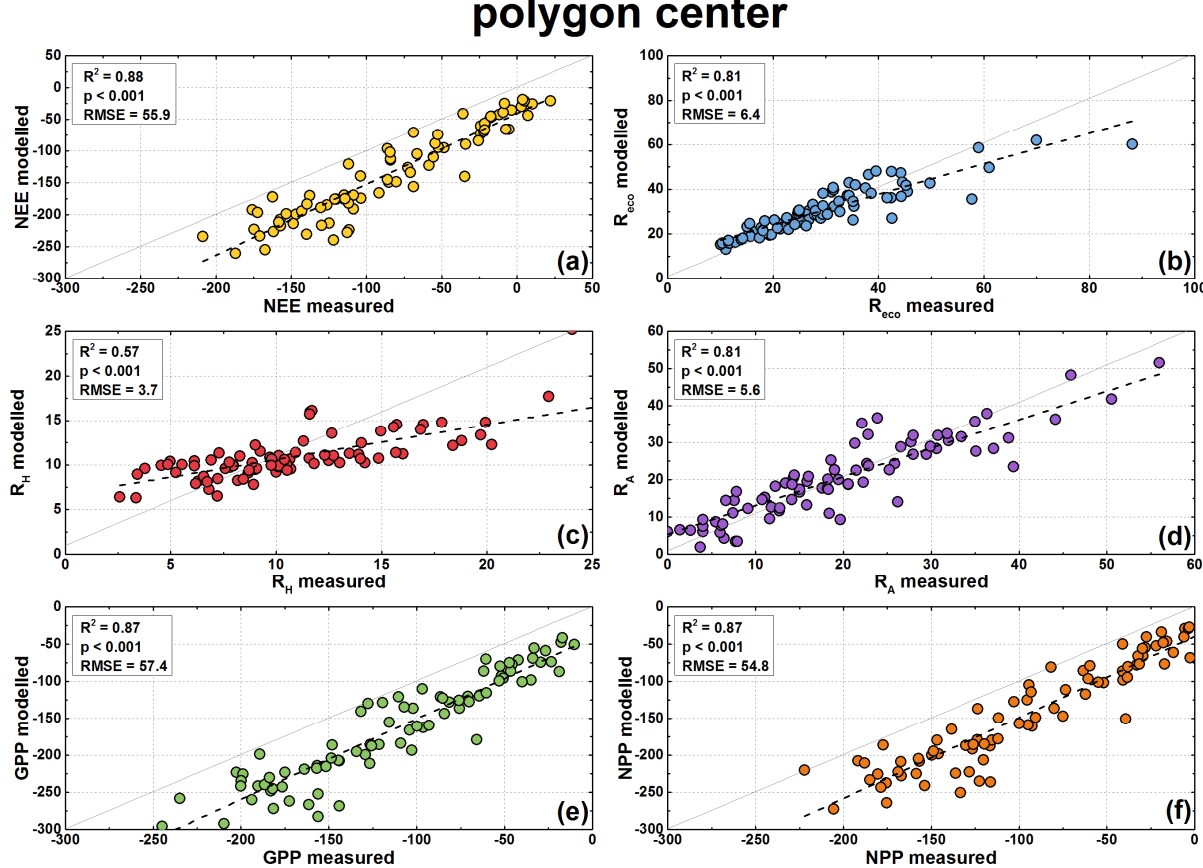

**Figure 6 - Modelled and measured CO$_2$ fluxes at the polygon center in µg CO$_2$ m$^{-2}$ s$^{-1}$. Measured fluxes are available for NEE (panel a), R$_{eco}$ (panel b) and R$_H$ (panel c). NEE model fluxes were calculated from modelled GPP (panel e) minus modelled R$_{eco}$, R$_A$ model fluxes (panel d) from modelled R$_{eco}$ minus modelled R$_H$ and NPP model fluxes (panel f) from modelled GPP minus modelled R$_A$. Note the different scales of the axes.**


# polygon rim

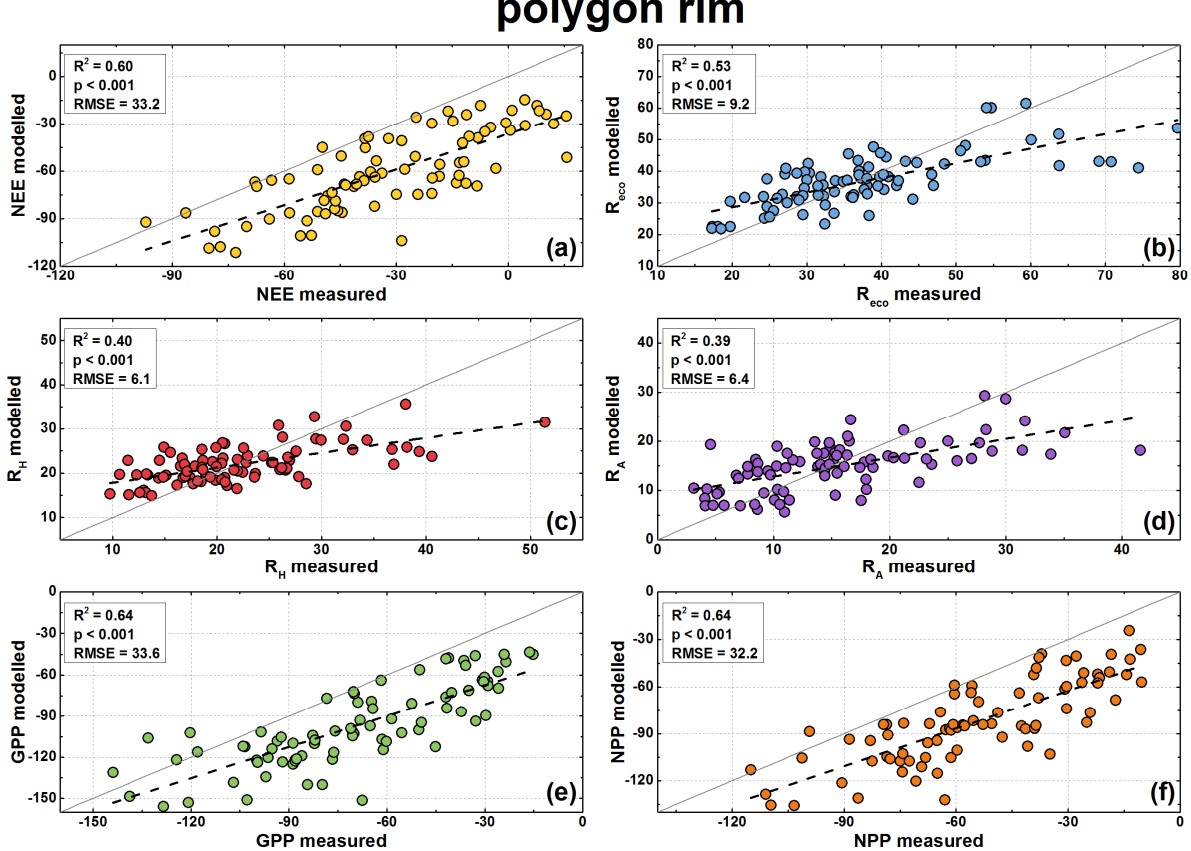

Figure 7 - **Modelled and measured CO₂ fluxes at the polygon rim in µg CO₂ m⁻² s⁻¹. Measured fluxes are available for NEE (panel a), R_eco (panel b) and R_H (panel c). NEE model fluxes were calculated from modelled GPP (panel e) minus modelled R_eco, R_A model fluxes (panel d) from modelled R_eco minus modelled R_H and NPP model fluxes (panel f) from modelled GPP minus modelled R_A. Note the different scales of the axes.**

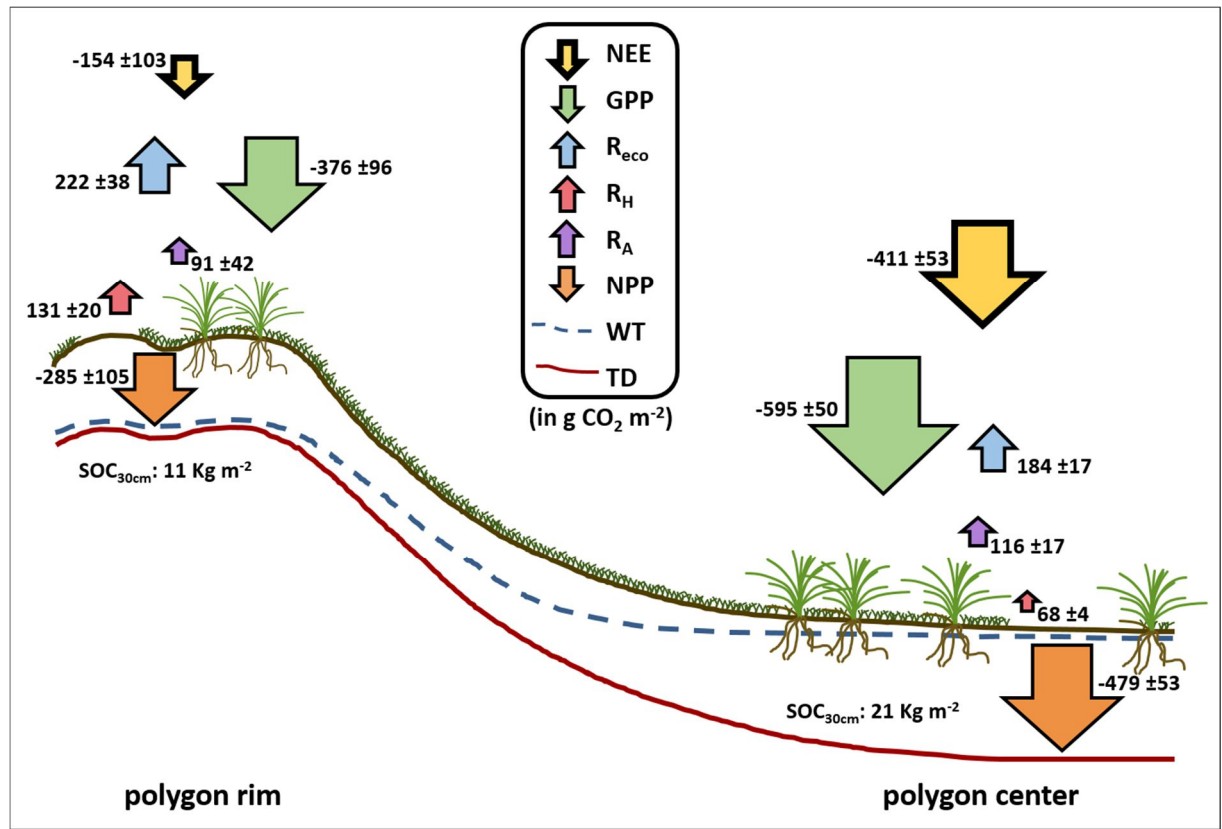

**Figure 8 – Integrated CO₂ fluxes at the polygon rim and center. The values were calculated from the model results and are given in g CO₂ m⁻². In total, both sites acted as a net CO₂ sink during the growing season. NEE= net ecosystem exchange; GPP= gross primary productivity; R_eco= ecosystem respiration; R_H= heterotrophic respiration; R_A= autotrophic respiration and NPP= net primary productivity; WT= water table; TD= thaw depth.**

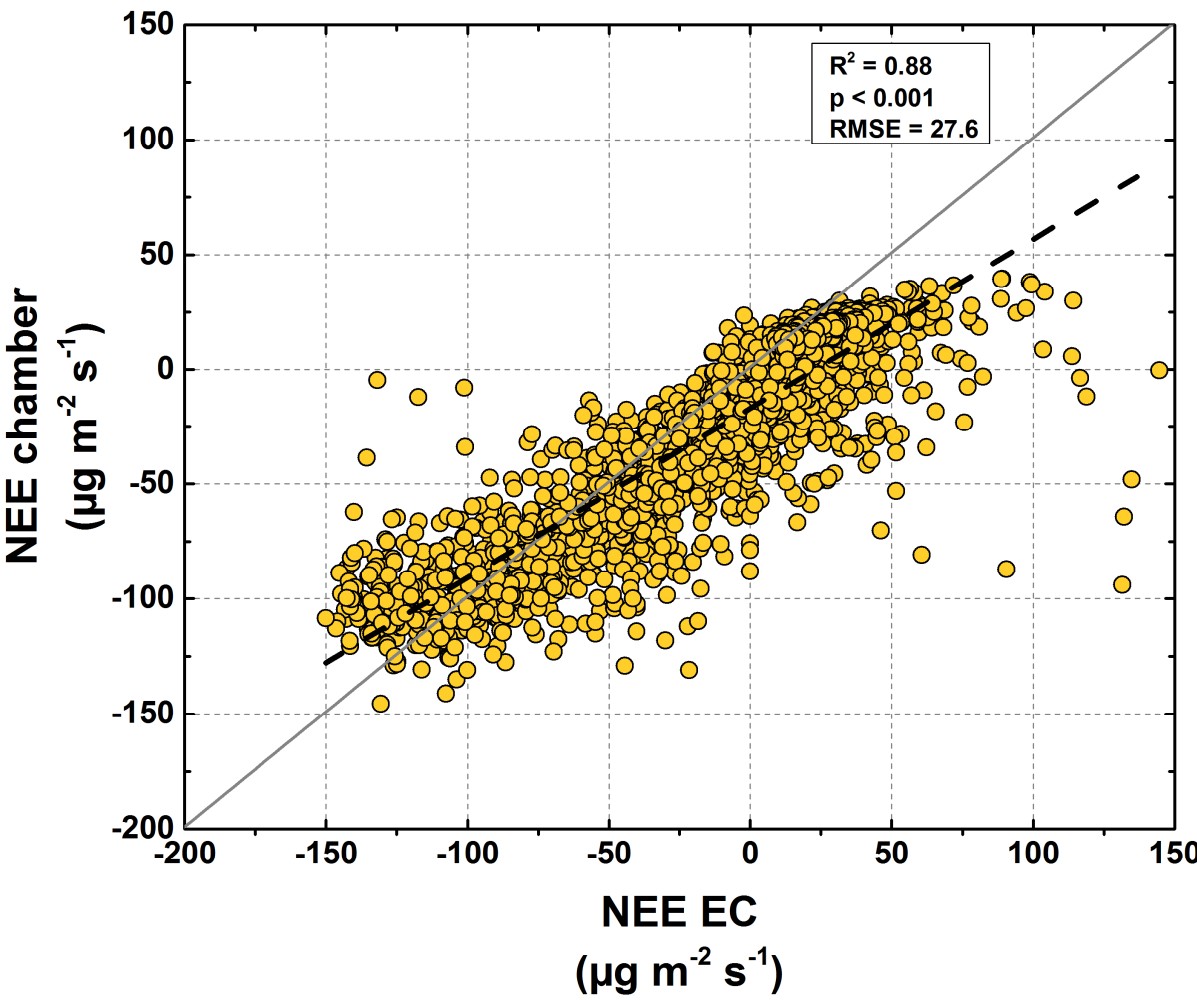

**Figure 9 – Comparison of chamber and half-hourly averaged EC NEE. The chamber NEE was calculated based on the contribution of each surface class to the EC footprint (equation 5).**