# Peer review of "Partitioning CO2 net ecosystem exchange fluxes on the pedon scale in the Lena River Delta, Siberia"

_Biogeosciences, 2018_

## Referee Comment (RC1) · A. J. Dolman (Referee) · 23 Oct 2018

The paper describes measurements of NEE, and the two respiratory fluxes on polygon tundra in the Siberian Arctic. They authors show that flux differences persist at the micro scale between the rim and the centre of the polygon. Although the work is generally okay, I think that there is somewhat of a missed opportunity here to use the eddy covariance data that are available for this site. As the authors say, the observations are well within the footprint of the EC system, so I am left wondering why these are not used to compare chamber NEE, or split to obtain EC GPP and Reco (line 116-119).

[Figure]

Can the authors explain why they do not use this data? Was it not available, or did it give different results (then it should certainly be used!).

Other comments.

L 33. Please be a little more precise. The Hugelius paper mentions 1300 Pg with an uncertainty range of 1100 to 1500 Pg.

L38. A more up to date reference about Arctic Amplification would be good. SWIPA 2017 would be appropriate.

L43. It would be appropriate to cite here Parmentier, et al., (2011).ÂăLonger growing seasons do not increase net carbon uptake in the northeastern Siberian tundra. Journal of Geophysical Research - Biogeosciences. ÂăISSNÂă2169-8953. Âă116(G04013).Âădoi: 10.1029/2011JG001653. Also because it is a site in the Siberian Arctic, as discussed below in l 44-55.

L 66. It may be better to refer to different sensitivity, rather than to "react", which is a result of the sensitivity.

L242. Fixing the Q10 is not necessarily the correct approach here. While it is difficult to estimate Rbase separately, just fixing it does not solve. It is important here to introduce the sensitivity to the definition of the Q10 as well as resulting uncertainty.

L275. This is really where I would have expected the use of the eddy covariance data.

L350 and Fig 6. I am not particularly impressed by the model-data comparison. It looks as if the fluxes are severely overestimated. Can the authors not provide a simple 1:1 scatterplot to show how well the model does?

L 280 and further. This section is very descriptive and basically repeats the graphics. It may be useful to see if and how far this can be reduced and made more concise. It does not really read nicely.

Table 2 could include the Parmentier paper mentioned earlier.

[Figure]

---

## Referee Comment (RC2) · Anonymous Referee #2 · 24 Oct 2018

Overall we recommend major revisions to highlight how the microsite-scale understanding from this study can extend to a better understanding of Arctic C flux dynamics.

General: This manuscript investigates effects of small-scale polygon heterogeneity on autotrophic and heterotrophic CO2 fluxes. The primary finding is that NEE spatial heterogeneity was very large, with four times more net CO2 uptake at polygon rims compared to centers. The CO2 flux rates varied with hydrology of the two rim locations, in part because GPP was higher and Rh lower in polygon centers compared to rims. The amount of information presented in the manuscript is impressive and the full partitioning of net CO2 fluxes into autotrophic and heterotrophic components provides

insight to mechanisms of spatial CO2 flux variation. The manuscript is based on an impressive dataset and would be improved by streamlining the results and crafting a stronger narrative to highlight the implications of these results for understanding Arctic C fluxes. The results should be shortened, and repetition removed. A number of environmental details could be condensed, for example by showing daily averages that are more relevant to the scale of sampling and highlighting only the model output that adds understanding to the measured data, like relevant physiological parameters or cumulative flux estimates. The discussion should consider the implications of these small-scale dynamics for understanding Arctic CO2 fluxes. Table 2 is an attempt to provide this context however the comparison to other sites across the Arctic seems anecdotal and raises more questions than it answers. Instead, the authors might consider relating the small-scale heterogeneity to net CO2 flux dynamics measured at the scale of flux towers, commenting on the relative balance of wet/dry sites across the island, and expected future trajectories for the island/region. It might also be interesting to discuss the role of water table versus plant biomass or other physiological drivers of C balance. Figure 8 is a nice summary and could make an even greater statement about the ecosystem C balance by incorporating the soil C estimates and literature-based plant biomass. More details are provided below.

Abstract Line 21-22: 'Fluxes measured at the microscale were used to model NEE, GPP, Reco, RH, RA and NPP over the growing season.' Modeled at what scale? It's a little unclear whether the fluxes were scaled up to a larger area or to get cumulative growing season estimates.

Line 22: 'For the first time' – first time ever in all permafrost systems? Or for the Lena River Delta?

Line 31: 'lad' should be led

Line 31: It would be helpful to conclude the abstract with a few words on the implications of the work.

Introduction Since this manuscript focuses on wet vs dry microsites the introduction should guide the reader toward moisture effects on CO2 flux, and interactions between moisture and warming. As it stands, the introduction focuses overwhelmingly on warming responses, partly because there is more literature on warming effects which is in itself a useful thing to highlight.

Line 43: There may be more appropriate citations here that specifically address plant and nutrient responses. For example: (Elmendorf et al. 2012, Salmon et al. 2016)

Line 46: It would be useful to be a little more specific with this statement. There are a number of studies that suggest the annual CO2 budget of arctic tundra is a weak sink to source (Oechel et al. 2014, Celis et al. 2017, Euskirchen et al. 2017) but that there's substantial spatial variation that we don't fully understand (Belshe et al. 2013, Ueyama et al. 2013). The effects of shifting hydrology are also not well understood.

Line 47: see also (McGuire et al. 2018)

Line 59: The discussion of variation in total flux magnitude could be condensed in this paragraph. The uncertainty related to hydrologic changes should be discussed.

Line 64: specify: 'inorganic fluxes are minor in highly organic soils'

Line 66-67: state briefly why it's important that the component fluxes react differently to changing conditions

Line 85-87: This sentence is very dense and so specific that it doesn't sufficiently highlight the uncertainties. The phrasing is also a little confusing because an increase in Ra would lead to a relative decrease in Rh but not necessarily an absolute decrease in Rh. And that detail isn't necessarily essential to the introduction. It would be helpful to discuss a little more generally how warming and moisture interact and highlight some of the competing CO2 flux processes. For example: warming stimulates plant productivity and CO2 uptake while increasing moisture has been found to suppress or stimulate both GPP and Reco (Chivers et al. 2009, Zona et al. 2012, Mauritz et

al. 2017). Drainage and warmer surface soils could reduce microbial biomass (Frey et al. 2008) however the effects could vary throughout the soil profile with drainage potentially stimulating decomposition of deeper soil C (Natali et al. 2015).

Line 86-87: (Segal and Sullivan 2014) might be a helpful citation regarding the contributions of root/shoot respiration and Rh to Reco.

Study Site Line 101: delete 'of' in 'depths of down to 300 to 500m'

Methods Line 185: Heterotrophic respiration section: The discussion of trenching and isotope methods producing relatively similar estimates of Rh might be better placed here than in the introduction. The introduction can then instead focus more on the big picture and include less methodological detail. This is a useful approach for fitting and evaluating NEE and Reco chamber measurements.

Line 193-196: what exactly does this 2014-2015 trenching comparison test?

Line 216: what is meant by 'the flux curve was re-inspected to see if irregularities could be removed by adjusting the time series'? What gets adjusted?

Line 240-245: Does this mean the only flexible and estimable parameter was Rbase?

Results Throughout, specify figure panels, eg: line 280 soil temperature (figure 2a).

Line 278 – 279: This sentence is out of place since it's a rim/center comparison and the following descriptions are all seasonal. The logical flow would be nicer with a general seasonal description followed by a microsite comparison.

Line 286: how does total precip compare to longer-term means?

Line 293-296: Is this level of detail on PAR necessary? It is impossible to see this detail in the figure, and the measurements were taken every few days so the detailed diurnal variation is less important. The occurrence of polar day/night is important and was already mentioned in the methods. A figure of daily PAR might be more useful since it would presumably show the declining light conditions toward the end of the season.

This high-resolution figure could go in the supplement, if it's necessary to refer to it at some point.

Line 299-306: This information is given in the site description, and it is unclear whether it's considered a result from the study or whether this data was collected simply for greater site characterization. Collecting this information is a lot of work and the details could be retained and moved to a supplement, perhaps with depth-resolved figures or tables which provide added value to the data from this paper but are not central to the results.

Line 300: a reduction in %C with depth at both the center and rim? Is the reduction in depth similar or do they reduce by different amounts?

Line 308: Start with the larger picture to put the fluxes in context. It's much more interesting and easier to read a description of the magnitudes and patterns of NEE, GPP, Reco, Ra, Rh and differences between microsites. Which microsite has higher sink strength? How do seasonal NEE patterns differ between center and rim? How do the magnitudes of Reco and GPP compare between center and rim? Does one site have more seasonal variation than the other? The specific max or min values or periods only need to be highlighted if it serves to illustrate something important or remarkable.

Line 346: The water analysis deserves its own section. What about correlations between VWC and R fluxes on the rim?

Line 351: Remind the reader what the parameters represent or refer back to the equations.

Line 354: This sentence says that Pmax showed strong temporal variation at the polygon center (mean 250.7 +/- 101.9) what does the +/- represent? Spatial variation around the mean? Or temporal variation? Is it a range, standard error, standard deviation, confidence interval?

[Figure]

Line 355: This might not be the most informative comparison given the very different temporal patterns in Pmax. In Figure 5b it looks like the patterns differ between Rim and Center until mid-August and then converge. That matches the GPP pattern between the two sites, and interestingly it does not coincide with marked changes in temperature or moisture. Perhaps it does coincide with the onset of nights?

Line 364: Hm, it's interesting that center is fit better with surface temperatures. Could this be related to the low fluctuation in soil temperature and the fact that surface temperature captures some of the variation in Reco that is related to Ra?

Line 368: averaged or cumulative? Why compare means instead of cumulatives?

Line 368 -397: This section is confusing, it repeats many of the flux results described above. It is unclear what additional information is gained from this detailed description of modeled fluxes. What do we learn from the means of the modeled fluxes? Isn't the main purpose of modeling to calculate seasonal cumulative fluxes?

Line 399: The previous section can be reduced, with far less detailed description of the modeled flux fluctuations. That space can be used to expand upon this section because it's very interesting. Address each flux component in turn, and how they compare between the two sites, and what that means for the NEE of each site.

Discussion: Line 406: This is a nice study with results that are a valuable contribution in their own right. Saying 'this is the first' doesn't necessarily elevate the results. Instead the value of the results might be better emphasized by highlighting the general differences in environmental conditions and fluxes between center and rim, and the most interesting elements of the results (like the different GPP:Reco ratios).

Line 412-414: That is interesting. That should definitely be more visible in the presentation of the results.

Line 421: starting the sentence with something other than 'Solely' would be better.

Line 421-423: Out of how many studies compared? Are these all the known studies

from Polygonal tundra? Based on (Virkkala et al. 2017)? And 3/8 studies agreeing means that about half the sites show comparable Reco.

Line 430: this section is misnamed since the majority of the writing is not about environmental controls. Environmental controls are typically abiotic factors and a lot of what is discussed here are vegetation factors.

Line 454-455: lead this paragraph with Reco or Rh since they are directly related to SOM decomposition.

Line 467-468: remain consistent in terminology rather than switching between NEE and net CO2 uptake.

Line 466-469: These trends are not terribly convincing. It is possible that the eye sees declining NEE in the center because of the steep slope from June to September and a smaller decline on the rim because NEE is overall lower through the season. What is the main argument here?

Line 481: What about (Dorrepaal et al. 2009, Schuur et al. 2009, Nowinski et al. 2010, Hicks Pries et al. 2013)?

Line 481: Unclear what 'these estimates of Rh' refers to. The previously cited studies? The results of this study?

Line 515: what is meant by recycled? The CO2 is taken up from the water column by plants before it can escape into the atmosphere? Is the argument here that declining Ra and Reco with rising water table is actually the result of CO2 uptake from the water column and thus a lower flux of CO2 to the atmosphere?

Line 528 – 532: This would be a useful statement in the introduction too.

Line 541: Except that Ra might not actually be driven by WT? Because the Ra measurement might in fact be affected by CO2 recycling? And the center vs rim comparison certainly does not suggest lower Ra in wet areas.

Figures and Tables Table 2: This table is not particularly helpful since it is unclear whether this is an exhaustive summary of other locations, or how this site relates to these other studies.

Figure 1: Turn landsat website into a citation so that the link can be removed from the caption. Just to make the caption a little cleaner.

Figure 2: This figure is difficult to read because of so much overlapping data within single panels. It should be revised to highlight only the most important variables, group variables with more logic (for example why is soil temperature in the panel with precipitation and air temperature in a separate panel (c)? it might make more sense to pair air temperature with precipitation). Consider showing these data at a temporal frequency more relevant to the measurements. panel b, add a line at y=0 to make it easier to see the WT relative to the soil surface. panel d give y-axis a negative scale otherwise it doesn't really make sense. At line 829 'rim an center' has a typo, fix to 'rim and center'

Figure 3: Add label for Polygon Center on the top and Polygon Rim on the bottom to make the figure easier to read at a glance.

Figure 4: panel letters are missing? Caption is incorrect in the flux sequence. For Rh and GPP, if the regressions are non-significant then there shouldn't be a line. Add a vertical line at 0cm to make it easier to see water table above and below the surface. Was this analysis done as a mixed effects model? Including a plot random effect might strengthen some of the relationships because it would control for plot-level variation (eg: biomass differences). Is this analysis picking up seasonal fluctuation in temperature (and light?) that coincides with rainfall and higher water tables. Even if the analysis is picking up seasonal variation in light and temperature RA and GPP would be expected to behave similarly. This is interesting to discuss.

Figure 5a: why isn't there an alpha parameter for center and rim sites?

Figure 6 & 7: move to supplement.

Figure 8: Nice way to summarise results! This figure would be easier to interpret if the arrows scaled by the size of the flux. It takes quite a lot of staring at the figure before it becomes clear that NEE is ∼3 times greater in the center. The figure could be even bolder by including C stock estimates for the soil and plants. Consider integrating the soil C profile data. Are there plant biomass estimates from other studies on Samoylov Island? It might get complicated but if it works then that would be a really nice synthesis of the C flux and partial C budgets for the two microsites. Add a label or legend item for the permafrost table and water table.

Belshe, E. F., E. A. G. Schuur, and B. M. Bolker. 2013. Tundra ecosystems observed to be CO2 sources due to differential amplification of the carbon cycle. Ecology Letters 16:1307–1315.

Celis, G., M. Mauritz, R. Bracho, V. G. Salmon, E. E. Webb, J. Hutchings, S. M. Natali, C. Schädel, K. G. Crummer, and E. A. G. Schuur. 2017. Tundra is a consistent source of CO2at a site with progressive permafrost thaw during 6 years of chamber and eddy covariance measurements. Journal of Geophysical Research: Biogeosciences 122:1471–1485.

Chivers, M. R., M. R. Turetsky, J. M. Waddington, J. W. Harden, and A. D. McGuire. 2009. Effects of Experimental Water Table and Temperature Manipulations on Ecosystem CO2 Fluxes in an Alaskan Rich Fen. Ecosystems 12:1329–1342.

Dorrepaal, E., S. Toet, R. S. P. Van Logtestijn, E. Swart, M. J. Van De Weg, T. V. Callaghan, and R. Aerts. 2009. Carbon respiration from subsurface peat accelerated by climate warming in the subarctic. Nature 460:616–619.

Elmendorf, S. C., G. H. R. Henry, R. D. Hollister, R. G. Bjork, N. Boulanger-Lapointe, E. J. Cooper, J. H. C. Cornelissen, T. A. Day, E. Dorrepaal, T. G. Elumeeva, M. Gill, W. A. Gould, J. Harte, D. S. Hik, A. Hofgaard, D. R. Johnson, J. F. Johnstone, I. S. Jonsdottir, J. C. Jorgenson, K. Klanderud, J. A. Klein, S. Koh, G. Kudo, M. Lara, E. Levesque, B. Magnusson, J. L. May, J. A. Mercado-Diaz, A. Michelsen, U. Molau, I. H.

Myers-Smith, S. F. Oberbauer, V. G. Onipchenko, C. Rixen, N. Martin Schmidt, G. R. Shaver, M. J. Spasojevic, ora E. orhallsdottir, A. Tolvanen, T. Troxler, C. E. Tweedie, S. Villareal, C.-H. Wahren, X. Walker, P. J. Webber, J. M. Welker, and S. Wipf. 2012. Plot-scale evidence of tundra vegetation change and links to recent summer warming. Nature Clim. Change 2:453–457.

Euskirchen, E. S., M. S. Bret-Harte, G. R. Shaver, C. W. Edgar, and V. E. Romanovsky. 2017. Long-Term Release of Carbon Dioxide from Arctic Tundra Ecosystems in Alaska. Ecosystems 20:960–974. Frey, S. D., R. Drijber, H. Smith, and J. Melillo. 2008. Microbial biomass, functional capacity, and community structure after 12 years of soil warming. Soil Biology and Biochemistry 40:2904–2907. Hicks Pries, C. E., E. A. G. Schuur, and K. G. Crummer. 2013. Thawing permafrost increases old soil and autotrophic respiration in tundra: Partitioning ecosystem respiration using $\delta$13C and $\Delta$14C. Global Change Biology 19:649–661.

Mauritz, M., R. Bracho, G. Celis, J. Hutchings, S. M. Natali, E. Pegoraro, V. G. Salmon, C. Schädel, E. E. Webb, and E. A. G. Schuur. 2017. Nonlinear CO2 flux response to 7 years of experimentally induced permafrost thaw. Global Change Biology 23:3646–3666.

McGuire, A. D., D. M. Lawrence, C. Koven, J. S. Clein, E. Burke, G. Chen, E. Jafarov, A. H. MacDougall, S. Marchenko, D. Nicolsky, S. Peng, A. Rinke, P. Ciais, I. Gouttevin, D. J. Hayes, D. Ji, G. Krinner, J. C. Moore, V. Romanovsky, C. Schädel, K. Schaefer, E. A. G. Schuur, and Q. Zhuang. 2018. Dependence of the evolution of carbon dynamics in the northern permafrost region on the trajectory of climate change. Proceedings of the National Academy of Sciences 115:3882–3887.

Natali, S. M., E. A. G. Schuur, M. Mauritz, J. D. Schade, G. Celis, K. G. Crummer, C. Johnston, J. Krapek, E. Pegoraro, V. G. Salmon, and E. E. Webb. 2015. Permafrost thaw and soil moisture driving CO2 and CH4 release from upland tundra. Journal of Geophysical Research: Biogeosciences 120:525–537.

Nowinski, N. S., L. Taneva, S. E. Trumbore, and J. M. Welker. 2010. Decomposition of old organic matter as a result of deeper active layers in a snow depth manipulation experiment. Oecologia 163:785–92.

Oechel, W. C., C. A. Laskowski, G. Burba, B. Gioli, and A. A. M. Kalhori. 2014. Annual patterns and budget of CO2 flux in an Arctic tussock tundra ecosystem. Journal of Geophysical Research: Biogeosciences 119:323–339.

Salmon, V. G., P. Soucy, M. Mauritz, G. Celis, S. M. Natali, M. C. Mack, and E. A. G. Schuur. 2016. Nitrogen availability increases in a tundra ecosystem during five years of experimental permafrost thaw. Global Change Biology 22:1927–1941.

Schuur, E. A. G., J. G. Vogel, K. G. Crummer, H. Lee, J. O. Sickman, and T. E. Osterkamp. 2009. The effect of permafrost thaw on old carbon release and net carbon exchange from tundra. Nature 459:556–559.

Segal, A. D., and P. F. Sullivan. 2014. Identifying the sources and uncertainties of ecosystem respiration in Arctic tussock tundra. Biogeochemistry 121:489–503.

Ueyama, M., H. Iwata, Y. Harazono, E. S. Euskirchen, W. C. Oechel, and D. Zona. 2013. Growing season and spatial variations of carbon fluxes of Arctic and boreal ecosystems in Alaska (USA). Ecological Applications 23:1798–1816.

Virkkala, A.-M., T. Virtanen, A. Lehtonen, J. Rinne, and M. Luoto. 2017. The current state of CO 2 flux chamber studies in the Arctic tundra: a review. Progress in Physical Geography 64:030913331774578.

Zona, D., D. A. Lipson, K. T. Paw U, S. F. Oberbauer, P. Olivas, B. Gioli, and W. C. Oechel. 2012. Increased CO 2 loss from vegetated drained lake tundra ecosystems due to flooding. Global Biogeochemical Cycles 26:1–16.

---

## Referee Comment (RC3) · Anonymous Referee #3 · 13 Nov 2018

The manuscript by Eckhardt et al., reports one growing season of CO2 flux data, not only NEE but its components GPP, RA, and RH, and their controlling factors in Lena Delta, Russia. It is extremely difficult to measure flux in such a remote area like Siberia and the result of this study will be highly valuable to flux community. Especially, measurement of in situ RA and RH is very rare especially in the Arctic region and this will be of great interest to readers of Biogeosciences. The manuscript is generally in good shape but several aspects should be addressed for the publication.

Comments: - Paragraph starting #78, warming effects on flux components are described in this paragraph but warming is not one of the main topics of this manuscript,

e.g. warming manipulation experiment. Thus, it does not seem appropriate for intro-
duction but rather for discussion that the results of this study imply xyz in the warming
scenario. - Line #82-4, if GEP is less sensitive to temperature than Reco, carbon sink
capacity will not be affected much by temperature instead of being reduced. Or car-
bon storage will be reduced because of a larger amount of C emission than C uptake.
Please rephrase it. - Paragraph starting #186, continuous regrowth of plants implies
living roots and remaining RA in the measured RH. In addition, if some roots are dying
after aboveground plant biomass is removed, can they add nutrients to soils and over-
estimate RH? It is written that there was no significant increase in RH, but continuous
and slow decay of remaining roots may affect RH. Also, was there any difference in the
plant regrowth rate between the center and the rim? If so, will they affect the results?
- Paragraphs starting #227, when modeling fluxes (Reco, RH, and GPP), some con-
stants (Q10, $\alpha$) were adopted from EC data. One of the purposes of this research is to
capture flux signals in microsite scale which EC cannot capture, and using constants
from EC data that contain a mixture of polygon centers and rims may decrease model
fit. Have you tried estimating Q10 and $\alpha$ with chamber flux data? It seems plausible to
estimate those values considering the number of data points. - Line #308-44, what are
the average values of NEE, Reco, GPP, and RH at the two microsites and how much
are those differences? These will be more important than the highest and the lowest
values, which took about half of this section space. - Line #325, RH seems corre-
lated with Reco, but no seasonal trend in RH was observed? At least RH in the center
seems to have seasonality in Figure 5. - Results of environmental controls on each flux
component is not described. Please add which environmental factors did or did not af-
fect flux components, which is one of the main objectives of this study. - Paragraph
starting #431, when discussing magnitude of fluxes and their explanatory factors, be
more specific if the difference is between Arctic ecosystems and other ecosystems in
the lower latitudes, or between this study site and other sites in the Arctic. - Line #454,
NEE → Reco? The following sentences are describing Reco and RH. In the separate
paragraph, the combined effects of GPP and Reco/RH can be described for NEE. -

Environmental controls on RA is not discussed.

---

## Author Response (AR1)

**We sincerely thank the reviewer for the comments, which helped us to substantially improve our manuscript. Please find the comments (black) and our reply (green) below.**

The paper describes measurements of NEE, and the two respiratory fluxes on polygon tundra in the Siberian Arctic. They authors show that flux differences persist at the micro scale between the rim and the centre of the polygon. Although the work is generally okay, I think that there is somewhat of a missed opportunity here to use the eddy covariance data that are available for this site. As the authors say, the observations are well within the footprint of the EC system, so I am left wondering why these are not used to compare chamber NEE, or split to obtain EC GPP and Reco (line 116-119). Can the authors explain why they do not use this data? Was it not available, or did it give different results (then it should certainly be used!).

By the time the manuscript was submitted, the EC dataset was not available. In the meantime the EC data were published (Holl et al., 2018) and are compared to the chamber data in the revised manuscript.

**Other comments**

L 33. Please be a little more precise. The Hugelius paper mentions 1300 Pg with an uncertainty range of 1100 to 1500 Pg.

Since we only refer to the organic carbon content in the uppermost three meters of permafrost affected soils (not the total organic carbon in the permafrost region) the number given by Hugelius et al. (2014) is  $1035 \pm 150$  Pg. We slightly modified the beginning of the sentence to 'About 1,000 Pg, which considers the uncertainty range.'

L38. A more up to date reference about Arctic Amplification would be good. SWIPA 2017 would be appropriate.

**We fully agree and added the suggested as well as another reference (Taylor et al., 2013).**

L43. It would be appropriate to cite here Parmentier, et al., (2011). Also because it is a site in the Siberian Arctic, as discussed below in I 44-55.

**We have added this reference here.**

L 66. It may be better to refer to different sensitivity, rather than to "react", which is a result of the sensitivity.

**We substantially revised the introduction and the mentioned sentence was re-written. Furthermore, we now use "respond" instead of "react" throughout the manuscript.**

L242. Fixing the Q10 is not necessarily the correct approach here. While it is difficult to estimate Rbase separately, just fixing it does not solve. It is important here to introduce the sensitivity to the definition of the Q10 as well as resulting uncertainty.

We have tried intensely to run the respiration models with a variable  $Q_{10}$  value. However, we decided to proceed with a fixed  $Q_{10}$  value because parameter estimation during fitting in MatLab did not converge to reasonable values for  $Q_{10}$  (around 1.5). We attribute this result to the relatively low number of samples available for fitting (about 150 samples per fitting) and to a tendency of the algorithm to overfit. The range of typical  $Q_{10}$  values of (soil) respiration has been shown to be rather narrow across different biomes with  $1.4 \pm 0.1$  (Mahecha et al., 2010). Moreover, following Runkle et al., (2013), for our site,  $Q_{10}$  has been estimated to lie within this range with  $1.5 \pm 0.3$  by Runkle et al. (2013) using eddy covariance data. We saw the availability of a site-specific  $Q_{10}$  as an opportunity to proceed with a less complex model. In an effort to avoid overfitting and emphasize parsimony we used prior process knowledge to reduce model complexity.

**L275. This is really where I would have expected the use of the eddy covariance data.**

We have now compared the modelled NEE chamber data with the eddy covariance data and the comparison showed good correlation. However, the modelled chamber NEE tended to underestimate the highest and lowest NEE in comparison to modelled EC NEE. Possible reasons for this bias is part of the discussion section.

L350 and Fig 6. I am not particularly impressed by the model-data comparison. It looks as if the fluxes are severely overestimated. Can the authors not provide a simple 1:1 scatterplot to show how well the model does?

**We replaced figures 6 and 7 with 1:1 scatterplots.**

L 280 and further. This section is very descriptive and basically repeats the graphics. It may be useful to see if and how far this can be reduced and made more concise. It does not really read nicely.

This comment is similar to those from the other two reviews. Therefore, we substantially revised the results section focusing on the most important results.

**Table 2 could include the Parmentier paper mentioned earlier.**

We have changed this table and put a focus solely on CO2 fluxes from either polygon rim or center microsites. Therefore, we have decided to not include the CO2 fluxes reported from Parmentier et al. (2011) as it presents CO2 fluxes of the polygonal tundra but not individual fluxes of rims and centres.

**Cited literature:**

- Holl, D., Wille, C., Sachs, T., Schreiber, P., Runkle, B. R. K., Beckebanze, L., Langer, M., Boike, J., Pfeiffer, E. M., Fedorova, I., Bolshiyanov, D., Grigoriev, M., and Kutzbach, L.: A long-term (2002 to 2017) record of closed-path and open-path eddy covariance CO2 net ecosystem exchange fluxes from the Siberian Arctic. 2018.
- Hugelius, G., Strauss, J., Zubrzycki, S., Harden, J. W., Schuur, E. A. G., Ping, C. L., Schirrmeister, L., Grosse, G., Michaelson, G. J., Koven, C. D., O'Donnell, J. A., Elberling, B., Mishra, U., Camill, P., Yu, Z., Palmtag, J., and Kuhry, P.: Estimated stocks of circumpolar permafrost carbon with quantified uncertainty ranges and identified data gaps, Biogeosciences, 11, 6573-6593, 2014.
- Mahecha, M. D., Reichstein, M., Carvalhais, N., Lasslop, G., Lange, H., Seneviratne, S. I., Vargas, R., Ammann, C., Arain, M. A., Cescatti, A., Janssens, I. A., Migliavacca, M., Montagnani, L., and Richardson, A. D.: Global Convergence in the Temperature Sensitivity of Respiration at Ecosystem Level, Science, 329, 838-840, 2010.
- Parmentier, F., Van Der Molen, M., Van Huissteden, J., Karsanaev, S., Kononov, A., Suzdalov, D., Maximov, T., and Dolman, A.: Longer growing seasons do not increase net carbon uptake in the northeastern Siberian tundra, Journal of Geophysical Research: Biogeosciences, 116, 2011.

- Runkle, B. R. K., Sachs, T., Wille, C., Pfeiffer, E. M., and Kutzbach, L.: Bulk partitioning the growing season net ecosystem exchange of CO2 in Siberian tundra reveals the seasonality of its carbon sequestration strength, Biogeosciences, 10, 1337-1349, 2013.
- Taylor, P. C., Cai, M., Hu, A., Meehl, J., Washington, W., and Zhang, G. J.: A decomposition of feedback contributions to polar warming amplification, Journal of Climate, 26, 7023-7043, 2013.

**We sincerely thank the reviewer for the comments, which helped us to substantially improve our manuscript. Please find the comments (black) and our reply (green) below.**

**General**

This manuscript investigates effects of small-scale polygon heterogeneity on autotrophic and heterotrophic CO2 fluxes. The primary finding is that NEE spatial heterogeneity was very large, with four times more net CO2 uptake at polygon rims compared to centers. The CO2 flux rates varied with hydrology of the two rim locations, in part because GPP was higher and Rh lower in polygon centers compared to rims. The amount of information presented in the manuscript is impressive and the full partitioning of net CO2 fluxes into autotrophic and heterotrophic components provides insight to mechanisms of spatial CO2 flux variation. The manuscript is based on an impressive dataset and would be improved by streamlining the results and crafting a stronger narrative to highlight the implications of these results for understanding Arctic C fluxes. The results should be shortened, and repetition removed. A number of environmental details could be condensed, for example by showing daily averages that are more relevant to the scale of sampling and highlighting only the model output that adds understanding to the measured data, like relevant physiological parameters or cumulative flux estimates. The discussion should consider the implications of these small-scale dynamics for understanding Arctic CO2 fluxes. Table 2 is an attempt to provide this context however the comparison to other sites across the Arctic seems anecdotal and raises more questions than it answers. Instead, the authors might consider relating the small-scale heterogeneity to net CO2 flux dynamics measured at the scale of flux towers, commenting on the relative balance of wet/dry sites across the island, and expected future trajectories for the island/region. It might also be interesting to discuss the role of water table versus plant biomass or other physiological drivers of C balance. Figure 8 is a nice summary and could make an even greater statement about the ecosystem C balance by incorporating the soil C estimates and literature-based plant biomass. More details are provided below.

Thank you for this general comment. We have substantially shortened the results section and removed repetitions to highlight the general differences of individual CO2 between rim and center. The section presenting environmental details was shortened and revised accordingly. We furthermore added a comparison between the measured chamber data and EC data from the same study site (Holl et al., 2018). The comparison of CO2 fluxes from this study with other chamber studies (Table 2) was substantially revised by focusing on chamber CO2 fluxes from polygonal tundra. Furthermore, Figure 8 was improved by adding estimates of soil C and scaling the size of the arrows based on the CO2 fluxes. Throughout the figures the colors of the single fluxes were synchronized.

**Abstract**

Line 21-22: 'Fluxes measured at the microscale were used to model NEE, GPP, Reco, RH, RA and NPP over the growing season.' Modeled at what scale? It's a little unclear whether the fluxes were scaled up to a larger area or to get cumulative growing season estimates.

We have revised this sentence to "The measured fluxes on the microscale (1 m - 10 m) were used to model the NEE, GPP,  $R_{eco}$ ,  $R_{H}$ ,  $R_{A}$  and net ecosystem production (NPP) to determine cumulative growing season fluxes".

Line 22: 'For the first time' – first time ever in all permafrost systems? Or for the Lena River Delta?

To the best of our knowledge this is the first time that the differing response of  $R_A$  and  $R_H CO_2$  fluxes to hydrological conditions have been examined in permafrost systems. We have revised this sentence accordingly.

**Line 31: 'lad' should be led**

**Changed accordingly.**

Line 31: It would be helpful to conclude the abstract with a few words on the implications of the work.

**We concluded the abstract with a summary of the implications of the current study.**

**Introduction**

Since this manuscript focuses on wet vs dry microsites the introduction should guide the reader toward moisture effects on CO2 flux, and interactions between moisture and warming. As it stands, the introduction focuses overwhelmingly on warming responses, partly because there is more literature on warming effects which is in itself a useful thing to highlight.

The introduction was substantially revised to consider to a greater extend changes in soil moisture in permafrost regions after warming and its effects on  $CO_2$  fluxes in arctic ecosystems.

Line 43: There may be more appropriate citations here that specifically address plant and nutrient responses. For example: (Elmendorf et al. 2012, Salmon et al. 2016)

**The respective literature is cited here.**

Line 46: It would be useful to be a little more specific with this statement. There are a number of studies that suggest the annual CO2 budget of arctic tundra is a weak sink to source (Oechel et al. 2014, Celis et al. 2017, Euskirchen et al. 2017) but that there's substantial spatial variation that we don't fully understand (Belshe et al. 2013, Ueyama et al. 2013). The effects of shifting hydrology are also not well understood.

Thank you for this important comment. In this part of the introduction we wanted to address the imbalance between the number of studies on  $CO_2$  fluxes from Russian and Alaskan tundra ecosystems. However, this part of the introduction was deleted. The introduction now focus on partitioning  $CO_2$  fluxes and the impact of environmental parameter on the individual fluxes rather than on  $CO_2$  budgets.

**Line 47: see also (McGuire et al. 2018)**

Since we made substantial changes within this paragraph the suggested reference does not fit anymore.

Line 59: The discussion of variation in total flux magnitude could be condensed in this paragraph. The uncertainty related to hydrologic changes should be discussed.

Thank you for this suggestion. We have made changes in the two sections between lines 44-67 focusing now on moisture effects on  $CO_2$  fluxes. This includes a substantial reduction of the discussion on the total flux magnitude.

Line 64: specify: 'inorganic fluxes are minor in highly organic soils'

We have revised the wording in the paragraph. Furthermore, we added values of total inorganic C (TIC) content in the last paragraph of section 4.1 as these values (just 0.2% TIC in all depths at rim and center) show the minor importance of TIC here.

Line 66-67: state briefly why it's important that the component fluxes react differently to changing conditions

A sentence on the importance of flux partitioning was added and embedded into the section about temperature and moisture impacts on  $CO_2$  fluxes.

Line 85-87: This sentence is very dense and so specific that it doesn't sufficiently highlight the uncertainties. The phrasing is also a little confusing because an increase in Ra would lead to a relative decrease in Rh but not necessarily an absolute decrease in Rh. And that detail isn't necessarily essential to the introduction. It would be helpful to discuss a little more generally how warming and moisture interact and highlight some of the competing CO2 flux processes. For example: warming stimulates plant productivity and CO2 uptake while increasing moisture has been found to suppress or stimulate both GPP and Reco (Chivers et al. 2009, Zona et al. 2012, Mauritz et al. 2017). Drainage and warmer surface soils could reduce microbial biomass (Frey et al. 2008) however the effects could vary throughout the soil profile with drainage potentially stimulating decomposition of deeper soil C (Natali et al. 2015).

Thank you for this helpful comment. Substantial changes were made in this part of the introduction to point out the importance of warming and changes of soil moisture on the individual CO2 fluxes. Here we have added the suggested research (Chivers et al. 2009, Zona et al. 2012, Natali et al. 2015, Mauritz et al. 2017)

Line 86-87: (Segal and Sullivan 2014) might be a helpful citation regarding the contributions of root/shoot respiration and Rh to Reco.

In the discussion section root/shoot respiration and  $R_h$  to  $R_{eco}$  is considered and the respective citation is now also considered in this part of the introduction.

**Study Site**

Line 101: delete 'of' in 'depths of down to 300 to 500m'

**Deleted.**

**Methods**

Line 185: Heterotrophic respiration section: The discussion of trenching and isotope methods producing relatively similar estimates of Rh might be better placed here than in the introduction. The introduction can then instead focus more on the big picture and include less methodological detail. This is a useful approach for fitting and evaluating NEE and Reco chamber measurements.

The discussion about methods to partition  $R_{eco}$  is included now into the method section 3.4 as suggested.

Line 193-196: what exactly does this 2014-2015 trenching comparison test?

The clipping and trenching method is related to considerable disturbances to the ecosystem. It was shown for other ecosystems that the additional decomposition of dead roots after clipping and trenching, can lead to an overestimation of  $R_H$  (Subke et al., 2006). Therefore, we compared  $CO_2$  fluxes from measurement plots that were trenched in 2014 with those that were trenched in 2015 to see if differences in  $R_H$  fluxes could be measured. We assumed that an additional decomposition of residual roots from plots trenched in 2014 would have ceased in 2015, one year after the treatment (following Diaz-Pines et al., 2010). The results have shown no significant differences between the plots that were trenched and clipped in different years. We have revised the respective section for clarity.

Line 216: what is meant by 'the flux curve was re-inspected to see if irregularities could be removed by adjusting the time series'? What gets adjusted?

The start time of the measurement was in some cases manually adjusted to remove irregularities of the flux curve. The start and end times were written down manually and were therefore partwise not identical to the real start of the flux measurement. We have revised this sentence and substituted 'adjusting the time series' with 'adjusting the flux calculation window'.

Line 240-245: Does this mean the only flexible and estimable parameter was Rbase?

Yes.

**Results**

Throughout, specify figure panels, eg: line 280 soil temperature (figure 2a).

**The figure panels were specified accordingly.**

Line 278 – 279: This sentence is out of place since it's a rim/center comparison and the following descriptions are all seasonal. The logical flow would be nicer with a general seasonal description followed by a microsite comparison.

To clearly distinguish between general seasonal descriptions and microsite comparison, we have placed the description of the soil temperature at the rim and center at the beginning of the next paragraph.

Line 286: how does total precip compare to longer-term means?

**We have added a comparison with precipitation data between 2003 and 2010.**

Line 293-296: Is this level of detail on PAR necessary? It is impossible to see this detail in the figure, and the measurements were taken every few days so the detailed diurnal variation is less important. The occurrence of polar day/night is important and was already mentioned in the methods. A figure of daily PAR might be more useful since it would presumably show the declining light conditions toward the end of the season. This high-resolution figure could go in the supplement, if it's necessary to refer to it at some point.

The complete figure was revised (see response below) and smaller adjustments were made to the text.

Line 299-306: This information is given in the site description, and it is unclear whether it's considered a result from the study or whether this data was collected simply for greater site characterization. Collecting this information is a lot of work and the details could be retained and moved to a supplement, perhaps with depth-resolved figures or tables which provide added value to the data from this paper but are not central to the results.

**We have shortened this paragraph and removed parts of the results as they are not central to the chamber flux results.**

Line 300: a reduction in %C with depth at both the center and rim? Is the reduction in depth similar or do they reduce by different amounts?

We have revised this sentence as the wording was not sufficient. The reduction of the total soil C content with depth was observed at both the center and rim, but more pronounced at the rim. Here the soil C content was half as much compared to the surface after 5 cm soil depth, while the C content at the center halved after 20 cm.

Line 308: Start with the larger picture to put the fluxes in context. It's much more interesting and easier to read a description of the magnitudes and patterns of NEE, GPP, Reco, Ra, Rh and differences between microsites. Which microsite has higher sink strength? How do seasonal NEE patterns differ between center and rim? How do the magnitudes of Reco and GPP compare between center and rim? Does one site have more seasonal variation than the other? The specific max or min values or periods only need to be highlighted if it serves to illustrate something important or remarkable.

We have revised the description of chamber flux results substantially and removed specific values. Instead, we put a focus on the description of the differences between the microsites and the seasonality of the single fluxes.

Line 346: The water analysis deserves its own section. What about correlations between VWC and R fluxes on the rim?

This is definitely one of the questions arising from the correlation of respiration fluxes with the water table at the polygon center. However, we haven't found a correlation between soil moisture and respiration fluxes at the polygon rim. Due to its elevation and the fast run-off of melt and precipitation water, the moisture regime at the polygon rim is completely different compared to the center. For instance, we discussed a 'recycling' of respired CO2 due to its slow diffusion through the moss layer (caused by a submersion of mosses), as possible reason for a correlation between RA fluxes and water table fluctuations at the polygon center. However, the moss layer at the polygon rim is not water-saturated and therefore respired CO2 can diffuse much faster through the moss layer than at the center. Furthermore, the moisture differences at the polygon rim are rather small, with a range between 28 and 34 % VWC, which might be not enough to cause differences in respiration fluxes. We added this to the discussion section 5.3.

Line 351: Remind the reader what the parameters represent or refer back to the equations.

We have added references to the equations in brackets.

Line 354: This sentence says that Pmax showed strong temporal variation at the polygon center (mean 250.7 +/- 101.9) what does the +/- represent? Spatial variation around the mean? Or temporal variation? Is it a range, standard error, standard deviation, confidence interval?

**It is a standard deviation of the daily averaged means and displays the temporal variation of the fitting parameter. We have revised the wording to clarify it.**

Line 355: This might not be the most informative comparison given the very different temporal patterns in Pmax. In Figure 5b it looks like the patterns differ between Rim and Center until mid-August and then converge. That matches the GPP pattern between the two sites, and interestingly it does not coincide with marked changes in temperature or moisture. Perhaps it does coincide with the onset of nights?

Thank you for this comment. Although  $P_{max}$  is strongly reduced at the onset of polar night the steep decrease in  $P_{max}$  at the polygon center is likely caused by plant senescence. Runkle et al. (2013) related the decrease of  $P_{max}$  at the end of August to plant senescence and we think that this factor leads to the convergence of the patterns between the two microsites. As discussed in section 5.2, the  $P_{max}$  at the polygon rim seems to be less affected by plant senescence, most probably due to the resilience of mosses, which are dominant at this site.

Line 364: Hm, it's interesting that center is fit better with surface temperatures. Could this be related to the low fluctuation in soil temperature and the fact that surface temperature captures some of the variation in Reco that is related to Ra?

Yes, we agree with the reviewer's interpretation. The higher sensitivity of  $R_{eco}$  fluxes at the polygon center to air/surface temperature is likely due to the sensitivity of  $R_A$  to changes in these temperature rather than changes in soil temperatures. At the polygon rim it is the other way round (the soil temperature describes the  $R_{eco}$  fluxes better than the surface temperature). This makes sense if the different contributions of  $R_H$  on  $R_{eco}$  fluxes are considered with contributions of >50% at the rim and <50% at the center. Giving this contribution, the Reco fluxes at the center are stronger affected by surface/air temperature as the fluxes are mainly driven by  $R_{A_r}$  while at the rim the fluxes are mainly driven by  $R_H$  and are therefore stronger affected by soil temperature. However, this holds not true for the modelled  $R_H$  fluxes as the  $R_H$  fluxes from the polygon center are better described by air than by soil temperature. Therefore, we cannot fully explain why the respiration fluxes are best described by air/surface temperatures at the polygon center. Both the soil temperature at polygon center and rim were measured at an adjacent polygon rim and center. The water table at the adjacent polygon center was permanently above the soil surface, while this was not the case at the polygon center where the flux measurements were conducted (see Fig 2). Therefore, there are most likely differences in soil temperatures in the upper soil layers between the polygon centers, which could lead to an attenuated fitting of the soil temperature with  $R_H$  fluxes at the center.

**Line 368: averaged or cumulative? Why compare means instead of cumulatives?**

**We do both a comparison of means and, later in the manuscript (section 4.4), a comparison of cumulative fluxes.**

Line 368 -397: This section is confusing, it repeats many of the flux results described above. It is unclear what additional information is gained from this detailed description of modeled fluxes. What

do we learn from the means of the modeled fluxes? Isn't the main purpose of modeling to calculate seasonal cumulative fluxes?

We have streamlined this section and put a focus on seasonal cumulative fluxes (section 4.4 and Fig. 8). However, in Table 1 we still show the mean values and ranges of the modelled fluxes as we think that especially the ranges are in particular cases of interest to the reader.

Line 399: The previous section can be reduced, with far less detailed description of the modeled flux fluctuations. That space can be used to expand upon this section because it's very interesting. Address each flux component in turn, and how they compare between the two sites, and what that means for the NEE of each site.

We have reduced the section 4.3 and adjusted 4.4 to show the differences of each flux component and their impact on differences in NEE fluxes between the microsites. As we discuss the impact of the single flux components on the net  $CO_2$  fluxes and their drivers intensely in the next section, we haven't expanded the mentioned section here.

**Discussion**

Line 406: This is a nice study with results that are a valuable contribution in their own right. Saying 'this is the first' doesn't necessarily elevate the results. Instead the value of the results might be better emphasized by highlighting the general differences in environmental conditions and fluxes between center and rim, and the most interesting elements of the results (like the different GPP:Reco ratios).

We revised this paragraph by highlighting the general differences between polygon rim and center.

Line 412-414: That is interesting. That should definitely be more visible in the presentation of the results.

We have added an additional sentence to highlight the differences in  $R_{eco}$  fluxes at the two sites in the results section 4.2.

Line 421: starting the sentence with something other than 'Solely' would be better.

This paragraph was revised substantially. We now focus on the comparison of  $CO_2$  fluxes from this study with other studies considering polygon rim and center microsites.

Line 421-423: Out of how many studies compared? Are these all the known studies from Polygonal tundra? Based on (Virkkala et al. 2017)? And 3/8 studies agreeing means that about half the sites show comparable Reco.

We have changed the comparison of  $CO_2$  fluxes from this study with other studies substantially. All the known chamber flux studies from polygon rims and centers are included (based on Virkkala et al., 2018).

Line 430: this section is misnamed since the majority of the writing is not about environmental controls. Environmental controls are typically abiotic factors and a lot of what is discussed here are vegetation factors.

We have changed the title of this section to 'Factors controlling CO2 fluxes'

Line 454-455: lead this paragraph with Reco or Rh since they are directly related to SOM decomposition.

**Changed accordingly.**

Line 467-468: remain consistent in terminology rather than switching between NEE and net CO2 uptake.

**We have harmonized the terminology throughout the complete manuscript and only use NEE.**

Line 466-469: These trends are not terribly convincing. It is possible that the eye sees declining NEE in the center because of the steep slope from June to September and a smaller decline on the rim because NEE is overall lower through the season. What is the main argument here?

We agree, that these trends are not convincing when considering the complete measurement period. However, by zooming into the fluxes during September, the trends are much clearer with a significant increase of NEE at the polygon rim and a slight decline of NEE at the polygon center. We discuss in section 5.2 that this increase might be assigned to the dense moss cover at the polygon rim, which might show low photosynthesis rates due to light stress during times of high PAR and desiccation (Murray et al., 1993, Zona et al., 2011). This interpretation is in accordance with the observation of rising NEE at the polygon rim when the drier period ended and PAR values were decreasing towards the end of the season (see Fig. 2).

Line 481: What about (Dorrepaal et al. 2009, Schuur et al. 2009, Nowinski et al. 2010, Hicks Pries et al. 2013)?

We didn't discuss the mentioned studies since they haven't estimated  $R_H$  fluxes over the growing season under in situ conditions. However, the wording was misleading and was revised to '(...) a few studies have estimated  $R_H$  fluxes from arctic tundra ecosystems over a growing season under in situ conditions'.

Line 481: Unclear what 'these estimates of Rh' refers to. The previously cited studies? The results of this study?

**We have revised the wording to '(...) differences in $R_H$ fluxes between these estimates and those presented in this study (...)'.**

Line 515: what is meant by recycled? The CO2 is taken up from the water column by plants before it can escape into the atmosphere? Is the argument here that declining Ra and Reco with rising water table is actually the result of CO2 uptake from the water column and thus a lower flux of CO2 to the atmosphere?

Yes, an uptake of CO2 from the water column by plants could serve as an explanation for the relationship between water table fluctuations and RA fluxes. The diffusion velocity through watersaturated soils is distinctly slower compared to well-aerated soils (Frank et al., 1996). Therefore, it seems plausible that a 'recycle' process as described by Liebner et al. (2011) gains more importance and lead to lower release of CO2 by RA. This process would affect Reco, not only RA fluxes. However, the relationship between RH fluxes and water table fluctuations might be missed due to the absence of photosynthetic active biomass in the measurement plots. We have revised this paragraph in section 5.3 substantially to clearly explain this effect on respiration fluxes. Line 528 – 532: This would be a useful statement in the introduction too.

Yes, while we have made substantial changes of the introduction with changing the focus from  $CO_2$ budgets towards a focus on the impact of environmental and vegetation factors on single  $CO_2$  fluxes, we have also added a sentence about the necessity of studying the impact of hydrological regimes on  $R_A$  fluxes.

Line 541: Except that Ra might not actually be driven by WT? Because the Ra measurement might in fact be affected by CO2 recycling? And the center vs rim comparison certainly does not suggest lower Ra in wet areas.

Yes, it might be possible that just the release of  $CO_2$  by  $R_A$  is driven by WT and not the  $R_A$  flux itself. Therefore, we revised this sentence accordingly. However, we think that there is a lower release of  $CO_2$  by  $R_A$  from the polygon center compared to the rim. Although the integrated fluxes are almost the same one has to consider the differences in GPP between the sites as photosynthesis is the source of  $R_A$ . The GPP: $R_A$  ratio at the polygon center is twice as high as at rim (10.5 and 5.1, respectively), which shows that about half as much  $CO_2$  is released by  $R_A$  at the center compared to the rim at similar GPP rates. These findings are added in section 5.3 to illustrate the difference in  $R_A$  fluxes from rim and center. Furthermore, the GPP and  $R_A$  fluxes at the rim are linearly correlated ( $R^2 = 0.48$ , p <0.05), with higher  $R_A$  during times of high GPP. This trend was not observed at the center ( $R^2 = 0.01$ , p > 0.05). This indicates that there certainly is a lower release of  $CO_2$  by  $R_A$  in wet areas.

**Figures and Tables**

Table 2: This table is not particularly helpful since it is unclear whether this is an exhaustive summary of other locations, or how this site relates to these other studies.

Thank you for this comment. We have made substantial changes to the comparison between  $CO_2$  fluxes from the current and previous studies on chamber  $CO_2$  fluxes from polygonal tundra sites. According to Virkkala et al., (2018),  $CO_2$  fluxes on the chamber scale (1 - 10 m) from polygonal tundra were only reported from Barrow and the Lena River Delta. We have also emphasized this fact in the introduction.

Figure 1: Turn landsat website into a citation so that the link can be removed from the caption. Just to make the caption a little cleaner.

**Changed.**

Figure 2: This figure is difficult to read because of so much overlapping data within single panels. It should be revised to highlight only the most important variables, group variables with more logic (for example why is soil temperature in the panel with precipitation and air temperature in a separate panel (c)? it might make more sense to pair air temperature with precipitation). Consider showing these data at a temporal frequency more relevant to the measurements. panel b, add a line at y=0 to make it easier to see the WT relative to the soil surface. panel d give y-axis a negative scale otherwise it doesn't really make sense.

We have revised the figure. We adjusted the temporal frequency to daily means instead of half-hourly means. Furthermore, we have added lines at y=0 if necessary and gave a negative scale for the panel with thaw depths. The precipitation data are now presented in an own panel.

At line 829 'rim an center' has a typo, fix to 'rim and center'

**Changed.**

Figure 3: Add label for Polygon Center on the top and Polygon Rim on the bottom to make the figure easier to read at a glance.

**We have added the labels. Furthermore, we have changed the colors of the single fluxes to synchronize them with the colors of the single fluxes of Fig. 8.**

Figure 4: panel letters are missing? Caption is incorrect in the flux sequence. For Rh and GPP, if the regressions are non-significant then there shouldn't be a line. Add a vertical line at 0cm to make it easier to see water table above and below the surface. Was this analysis done as a mixed effects model? Including a plot random effect might strengthen some of the relationships because it would control for plot-level variation (eg: biomass differences). Is this analysis picking up seasonal fluctuation in temperature (and light?) that coincides with rainfall and higher water tables. Even if the analysis is picking up seasonal variation in light and temperature RA and GPP would be expected to behave similarly. This is interesting to discuss.

Thank you for this comment. We have removed the regression line for insignificant fluxes and added a vertical line at 0 cm. Unfortunately, we were not able to control for plot-level variation as the RA fluxes were calculated from fluxes from different measurement plots and not measured directly. Furthermore, there are no estimates of biomass for each plot. The analysis might pick up seasonal variation in radiation and temperature, but we estimate that the effect on the regression itself is low as the water table reacts rather slow to variation of both temperature and light. Furthermore, we discuss the different behavior of GPP and RA fluxes to changes of the mentioned parameter in the section 5.3.

**Figure 5a: why isn't there an alpha parameter for center and rim sites?**

The values for the initial canopy quantum efficiency  $\alpha$  were obtained from modelled fluxes of the Eddy Covariance measurement system (Holl et al., 2018). The footprint of the EC system contains both polygon rims and centers. Therefore, the same value of the  $\alpha$  parameter was used for both microsites. We have added this in section 3.6.

**Figure 6 & 7: move to supplement.**

**According to the suggestions of reviewer #1, the presentation of these data were revised substantially but still included in the main part of the manuscript.**

Figure 8: Nice way to summarise results! This figure would be easier to interpret if the arrows scaled by the size of the flux. It takes quite a lot of staring at the figure before it becomes clear that NEE is  $\sim$  3 times greater in the center. The figure could be even bolder by including C stock estimates for the soil and plants. Consider integrating the soil C profile data. Are there plant biomass estimates from other studies on Samoylov Island? It might get complicated but if it works then that would be a really nice synthesis of the C flux and partial C budgets for the two microsites. Add a label or legend item for the permafrost table and water table.

The size of the arrows were scaled to the size of the flux and legend items for the water table and the thaw depth were added. Furthermore, we have added estimates of SOC for both microsites. Estimates

of aboveground biomass for both microsites are lacking. There are estimates for a polygon rim and center from Samoylov Island from the literature (see Zhang et al., 2012), but they differ distinctly from what we have found at the study site (e.g. total aboveground biomass is higher at the rim than at the center). Therefore, we have decided not to include these estimates here.

**Cited literature:**

- Chivers, M. R., Turetsky, M. R., Waddington, J. M., Harden, J. W., and McGuire, A. D.: Effects of Experimental Water Table and Temperature Manipulations on Ecosystem CO2 Fluxes in an Alaskan Rich Fen, Ecosystems, 12, 1329-1342, 2009.
- Diaz-Pines, E., Schindlbacher, A., Pfeffer, M., Jandl, R., Zechmeister-Boltenstern, S., and Rubio, A.: Root trenching: a useful tool to estimate autotrophic soil respiration? A case study in an Austrian mountain forest, European Journal of Forest Research, 129, 101-109, 2010.
- Frank, M. J., Kuipers, J. A., van Swaaij, W. P. J. J. o. C. and Data, E.: Diffusion coefficients and viscosities of CO2+ H2O, CO2+ CH3OH, NH3+ H2O, and NH3+ CH3OH liquid mixtures, Journal of Chemical & Engineering Data, 41, 297-302, 10.1021/je950157k, 1996.
- Holl, D., Wille, C., Sachs, T., Schreiber, P., Runkle, B. R. K., Beckebanze, L., Langer, M., Boike, J., Pfeiffer, E. M., Fedorova, I., Bolshiyanov, D., Grigoriev, M., and Kutzbach, L.: A long-term (2002 to 2017) record of closed-path and open-path eddy covariance CO2 net ecosystem exchange fluxes from the Siberian Arctic. 2018.
- Murray, K., Tenhunen, J., and Nowak, R.: Photoinhibition as a control on photosynthesis and production of Sphagnum mosses, Oecologia, 96, 200-207, 1993.
- Liebner, S., Zeyer, J., Wagner, D., Schubert, C., Pfeiffer, E.-M., and Knoblauch, C.: Methane oxidation associated with submerged brown mosses reduces methane emissions from Siberian polygonal tundra, Journal of Ecology, 99, 914-922, 2011.
- Runkle, B. R. K., Sachs, T., Wille, C., Pfeiffer, E. M., and Kutzbach, L.: Bulk partitioning the growing season net ecosystem exchange of CO2 in Siberian tundra reveals the seasonality of its carbon sequestration strength, Biogeosciences, 10, 1337-1349, 2013.
- Subke, J.-A., Inglima, I., and Francesca Cotrufo, M.: Trends and methodological impacts in soil CO2 efflux partitioning: A metaanalytical review, Global Change Biology, 12, 921-943, 2006.
- Virkkala, A. M., Virtanen, T., Lehtonen, A., Rinne, J., and Luoto, M.: The current state of CO2 flux chamber studies in the Arctic tundra: A review, Progress in Physical Geography, 42, 162-184, 2018.
- Zhang, Y., Sachs, T., Li, C. and Boike, J.: Upscaling methane fluxes from closed chambers to eddy covariance based on a permafrost biogeochemistry integrated model, Global Change Biology, 18, 1428-1440, doi:10.1111/j.1365-2486.2011.02587.x, 2012.
- Zona, D., Oechel, W. C., Richards, J. H., Hastings, S., Kopetz, I., Ikawa, H., and Oberbauer, S.: Light-stress avoidance mechanisms in a Sphagnum-dominated wet coastal Arctic tundra ecosystem in Alaska, Ecology, 92, 633-644, 2011.

**We sincerely thank the reviewer for the comments, which helped us to substantially improve our manuscript. Please find the comments (black) and our reply (green) below.**

The manuscript by Eckhardt et al., reports one growing season of CO2 flux data, not only NEE but its components GPP, RA, and RH, and their controlling factors in Lena Delta, Russia. It is extremely difficult to measure flux in such a remote area like Siberia and the result of this study will be highly valuable to flux community. Especially, measurement of in situ RA and RH is very rare especially in the Arctic region and this will be of great interest to readers of Biogeosciences. The manuscript is generally in good shape but several aspects should be addressed for the publication.

**Comments:**

Paragraph starting #78: warming effects on flux components are described in this paragraph but warming is not one of the main topics of this manuscript, e.g. warming manipulation experiment. Thus, it does not seem appropriate for introduction but rather for discussion that the results of this study imply xyz in the warming scenario.

Thank you for this comment. The introduction was substantially revised focusing rather on hydrology effects on  $CO_2$  fluxes, which are actually reported in the current manuscript (see also comments of reviewer #2). However, warming effects are still considered in the introduction since they also affect changes in hydrology, e.g. through permafrost thaw.

Line #82-4: if GEP is less sensitive to temperature than Reco, carbon sink capacity will not be affected much by temperature instead of being reduced. Or carbon storage will be reduced because of a larger amount of C emission than C uptake. Please rephrase it.

**We agree, the wording is misleading here. We have revised 'carbon sink capacity' to 'carbon storage'.**

Paragraph starting #186: continuous regrowth of plants implies living roots and remaining RA in the measured RH. In addition, if some roots are dying after aboveground plant biomass is removed, can they add nutrients to soils and overestimate RH? It is written that there was no significant increase in RH, but continuous and slow decay of remaining roots may affect RH. Also, was there any difference in the plant regrowth rate between the center and the rim? If so, will they affect the results?

We addressed this question with different approaches. The continuous re-growth of plants implies living roots and remaining  $R_A$  in the measured flux which we define as  $R_H$ . However, we expect minor effects of additional decay of dead roots and release of nutrients to the measured respiration fluxes. There was only a very sparse re-growth of plants at the measurement plots where we have removed the photosynthetic active biomass, so we assume that it was negligible for the flux measurements. We also haven't seen any differences in the amount of plant re-growth between rim and center plots. It is possible that nutrients were released to the soil due to dying roots and that the decay of dead roots lead to an overestimation of  $R_H$  fluxes. However, we have removed the biomass from plots in 2014 and from other plots in 2015 to see if there are effects due to dying roots and nutrient addition (see response to reviewer #2). A Student's t-test revealed no significant differences between plots that were manipulated in 2014 and 2015. The lack in a significant difference between  $R_H$  in the plots clipped in 2014 and 2015 means that either no significant amounts of  $CO_2$  from root biomass contribute to  $CO_2$  fluxes or that the  $CO_2$  release from decaying roots does not diminish over the period of one year, which seems unlikely. A lack of  $CO_2$  release from the clipped root biomass is also supported by a study of Biasi et al. (2014) who have compared the same partitioning approach with a non-disturbing 14C partitioning approach and found no significant differences in the measured  $R_H$  fluxes between the two approaches.

Paragraphs starting #227: when modeling fluxes (Reco, RH, and GPP), some constants (Q10, \_) were adopted from EC data. One of the purposes of this research is to capture flux signals in microsite scale which EC cannot capture, and using constants from EC data that contain a mixture of polygon centers and rims may decrease model fit. Have you tried estimating Q10 and \_ with chamber flux data? It seems plausible to estimate those values considering the number of data points.

We have tried intensively to run the models with solely chamber flux data as we also wanted to determine individual constants for polygon rim and center. However, parameter estimation during the fitting did not converge to reasonable values for  $Q_{10}$  when the fitting was made solely with chamber flux data (see response to reviewer #1). We attribute these findings to the relatively low number of samples available for fitting. Therefore, we have decided to run the models with site-specific constants obtained from EC data.

Line #308-44: what are the average values of NEE, Reco, GPP, and RH at the two microsites and how much are those differences? These will be more important than the highest and the lowest values, which took about half of this section space.

Thank you for this comment. We have substantially revised this section and decided, to forgo to show specific values of chamber fluxes (see response to reviewer #2). Instead, the differences between the microsites and the seasonality of the single fluxes were highlighted.

Line #325: RH seems correlated with Reco, but no seasonal trend in RH was observed? At least RH in the center seems to have seasonality in Figure 5. - Results of environmental controls on each flux component is not described. Please add which environmental factors did or did not affect flux components, which is one of the main objectives of this study.

There might be a slight seasonal trend of  $R_{eco}$  fluxes at the polygon rim, which may be also seen in the  $R_{H}$  fluxes from this microsite (see Fig. 5). However, at the polygon center no seasonality is seen for  $R_{H}$  fluxes (open symbols in panel d of Fig. 5 in original manuscript). We also expected a trend in the contribution of  $R_{H}$  on  $R_{eco}$  due to plant senescence and root mortality at the end of the growing season. However, neither at the rim nor at the center a seasonal trend of this contribution was observed. This is in contrast to the study from Segal and Sullivan (2014) where the contribution of  $R_{H}$  increased towards the end of the growing season, most likely due to deepening of the active layer which increases substrate availability for  $R_{H}$  production processes. This effect might be missed in this study because of smaller changes in thaw depth as well as lower soil temperatures throughout the growing season at the study site compared to other arctic tundra sites and due to a too short investigation period. The main environmental drivers of the  $CO_2$  fluxes are PAR for GPP fluxes and the temperature for respiration fluxes (see Fig. 6 & 7 in revised manuscript). Furthermore, the hydrology is a main driver of the respiration fluxes, especially  $R_{eco}$  and  $R_A$  fluxes (see panel a & d of Fig. 4 in revised manuscript). These relationships have been shown with the good fitting of the flux models.

Paragraph starting #431: when discussing magnitude of fluxes and their explanatory factors, be more specific if the difference is between Arctic ecosystems and other ecosystems in the lower latitudes, or between this study site and other sites in the Arctic.

**We have revised this paragraph accordingly and clarified, which ecosystems are compared.**

Line #454: NEE ! Reco? The following sentences are describing Reco and RH. In the separate paragraph, the combined effects of GPP and Reco/RH can be described for NEE. -Environmental controls on RA is not discussed.

Thank you for this comment. We have changed NEE into  $R_{eco}$ , which is the right term here. The combined effects are discussed, as suggested, in a separate paragraph. The environmental controls of  $R_A$  fluxes are also discussed, but later in section 5.3.

**Cited literature:**

- Biasi, C., Jokinen, S., Marushchak, M. E., Hämäläinen, K., Trubnikova, T., Oinonen, M., and Martikainen,
   P. J.: Microbial Respiration in Arctic Upland and Peat Soils as a Source of Atmospheric Carbon
   Dioxide, Ecosystems, 17, 112-126, 2014.
- Segal, A. D. and Sullivan, P. F.: Identifying the sources and uncertainties of ecosystem respiration in Arctic tussock tundra, Biogeochemistry, 121, 489-503, 2014.

**Please find below a list of all relevant changes made in the manuscript**

- We streamlined the introduction section towards a focus on moisture effects on individual CO2 fluxes in arctic tundra landscapes
- We added a comparison of the modelled chamber data with the available eddy covariance data:
  - The calculation and weighting of the fluxes for the comparison is described in the methods section 3.6
  - The results of the comparison are described in section 4.4 and diagrammed in figure
     9, which we added to the manuscript.
- We intensely revised the results sections 4.1 to put a focus simply on relevant data on relevant time scales
- We intensely revised the results sections 4.2 and 4.3 by removing the listings of individual CO2 flux values as all relevant values are listed in the tables 1 and 2. Instead, we put a focus on describing differences of the individual CO2 fluxes between the microsites.
- We revised the complete discussion section intensely based on the comments of the three reviewer.
- Table 2 was modified to put the focus on chamber CO2 fluxes from other polygonal tundra microsites as well as other wet and dry microsites of arctic tundra ecosystems. Furthermore, we have rewritten the discussion section 5.2 where the comparison of CO2 chamber fluxes is discussed.
- The discussion section 5.3 was intensely revised, based on the comments of review #2, as some ideas on autotrophic respiration fluxes were not conclusive.
- Throughout the figures, we have synchronized the colors codes of the individual CO2 fluxes.
- We modified figure 3. Instead of half-hourly mean values daily mean values are presented.
- We replaced the figures 6 and 7 with 1:1 scatterplots of modelled and measured CO2 flux data.
- We deposited the used data of the current manuscript on PANGAEA (https://doi.pangaea.de/10.1594/PANGAEA.898876) and added this reference
- We improved the style of figure 8 (scaled arrows) and added information on carbon stocks.

**Partitioning CO2 net ecosystem exchange fluxes on the microsite pedon scale in the Lena River Delta, Siberia**

Tim Eckhardt1,2, Christian Knoblauch1,2, Lars Kutzbach1,2, David Holl1,2, Gillian Simpson3, Evgeny Abakumov4 and Eva-Maria Pfeiffer1,2

1Institute of Soil Science, Universität Hamburg, Allende-Platz 2 Hamburg, 20146, Germany
 2Center for Earth System Research and Sustainability, Universität Hamburg, Allende-Platz 2, Hamburg, 20146, Germany
 3School of GeoSciences, University of Edinburgh, West Mains Road, Edinburgh, EH9 3JN, Scotland, UK

4Department of Applied Ecology, Saint-Petersburg State University, 199178, 16-line 2, Vasilyevskiy Island,

10 Russia

Correspondence to: Tim Eckhardt (tim.eckhardt@uni-hamburg.de)

**Abstract.** Arctic tundra ecosystems are currently facing amplified rates of amplified climate changewarming. This is critical as theseSince these ecosystems store significant amounts of soil organic carbon-in their soils, which can be mineralized to carbon dioxide (CO2-) and methane (CH4-), rising temperatures may cause increasing greenhouse

- 15 gas fluxes to the atmosphere and released to the atmosphere. To understand how the CO2 net ecosystem exchange (NEE) fluxes will reactrespond to changing climatic and environmental conditions, it is necessary to understand the individual responses of the physiological processes contributing to CO2 NEE. Therefore, this study aimed: i) to partition NEE fluxes at the soil-plant-atmosphere interface in an arctic tundra ecosystem; and ii) to identify the main environmental drivers of these fluxes. Hereby, the NEE fluxes were partitioned into gross primary
- 20 productivity (GPP) and ecosystem respiration (Reco) and further into autotrophic (RA) and heterotrophic respiration (RH). The study examined CO2 flux data collected during the growing season in 2015 using closed chamber measurements in a polygonal tundra landscape in the Lena River Delta, northeastern Siberia. To capture the influence of soil hydrology on CO2 fluxes, measurements were conducted at a water-saturated polygon center and a well-drained polygon rim. These chamber-measured fluxes on the microscale (1 m 10 m)-were used to model
- 25 the NEE, GPP,  $R_{eco}$ ,  $R_H$ ,  $R_A$  and net ecosystem primary production (NPP) at the pedon scale (1 m 10 m) over the growing seasonand to determine cumulative growing season fluxes. Here, for the first time, the differing response of *in situ* measured  $R_A$  and  $R_H$  fluxes from permafrost-affected soils to hydrological conditions have been examined. Although changes in the water table depth at the polygon center sites did not affect CO2 fluxes from  $R_H$ , rising water tables were linked to reduced CO2 fluxes from  $R_{A_c}$ -It was shown that low  $R_A$ -fluxes are associated
- 30 to a high water table, most likely due to the submersion of mosses, while an effect of water table fluctuations on RH fluxes was not observed. Furthermore, this work found the polygonal tundra in the Lena River Delta to be a net\_sink for atmospheric CO2 during the growing season. Spatial heterogeneity was apparent with the net CO2 uptakeThe NEE at a the wet, depressed polygon center being-was more than twice as high as that measured than at a the drier polygon rim. In addition to higher GPP fluxes, these differences in NEE between the two microsites
- 35 were caused by higher GPP fluxes, due to a higher vascular plant density and lower Reco fluxes due to oxygen limitation under water-saturated conditions at the polygon center in comparison to the rim. lower Reco fluxes at the center compared to the rim. Here, the contrasting-Hence, soil hydrological conditions were one of the key drivers for the different CO2 fluxes across this highly heterogeneous tundra landscape. caused the CO2 flux differences between the microsites, where high water levels lad to lower decomposition rates due to anoxic conditions.

**40 1 Introduction**

More than An estimated 1,000 Petagrams of organic carbon (OC) are stored in the upper 3 m of northern permafrost-affected soils (Hugelius et al., 2014). Given the large amount of OC stored in these soils, the response of the arctic carbon (C) cycle to a changing climate is of global importance (McGuire et al., 2009). Over thousands of years, Ccarbon has been sequestered in permafrost-affected soils and sediments due to cold conditions and poor drainage resulting in water saturation and slow organic matter decomposition. Currently, arctic ecosystems are

- drainage resulting in water saturation and slow organic matter decomposition. Currently, arctic ecosystems are facing amplified warming (AMAP, 2017; Taylor et al., 2013), which will lead to the longer and deeper thawing of permafrost-affected soils (Romanovsky et al., 2010). On the one hand, the microbial decomposition of newly available liberated-thawed permafrost organic matter releases carbon dioxide (CO2) and methane (CH4) (e.g. Knoblauch et al., 2018; Knoblauch et al., 2013; Zimov et al., 2006a; Schuur et al., 2009; Grosse et al., 2011). On the other hand, higher temperatures increase the assimilation of CO2 by tundra vegetation due to a longer-prolonged
- growing period and increased nutrient availability in the deeper layers of thawed soils (e.g. Beermann et al., 2017; Elmendorf et al., 2012; Salmon et al., 2016; Parmentier et al., 2011).

Although the CO2 budget of the arctic tundra has been the topic of several studies (Euskirchen et al., 2017; Ueyama et al., 2013; Merbold et al., 2009; e.g. Kittler et al., 2016; Marushchak et al., 2013; Oechel et al., 2000;

[revised manuscript text omitted]
 were found at the polygon center in the entirethroughout the active layer (Zubrzycki et al., 2013). The vV egetation of theon polygon rims is dominated by mosses (*Hylocomium splendens, Polytrichum spp., Rhytidium rugosum*), some small vascular plants (*Dryas punctata* and *Astragalus frigidus*) as well as lichens (*Peltigera spp.*) and was can be classified as non-tussock sedge, dwarf-shrub, moss tundra (after (Walker et al., 2005)) as non-tussock sedge.
- 185 dwarf-shrub, moss tundra. The vegetation of the polygon centers were dominated by the hydrophilic sedge *Carex* aquatilis, which have in general much higher growth forms than at the rim, and mosses (*Drepanocladus revolvens*, *Meesia triqueta, Scorpidium scorpioides*) and classified as sedge, moss, dwarf-shrub wetland according to (Walker et al. (2005).

**3** Methods**

195

**190 **3.1 Meteorological data**

Meteorological variables were recorded at 30 minute intervals at the nearby EC system and adjacent meteorological station 40 m southwest of the study site. Data were collected on relative humidity and air temperature (MP103A, ROTRONIC AG, Switzerland), air pressure (RPT410F, Druck Messtechnik GmbH, Germany) and photosynthetically active radiation (PAR; wavelength: 400 – 700 nanometers; QS2, Delta-T Devices Ltd., UK) as well as the incoming and reflected components of shortwave and longwave radiation, respectively (CNR 1, Kipp and Zonen, Netherlands)<del>, were collected</del>. The radiative surface temperature (Tsurf, in Kelvin (K)) was calculated as:

$$T_{surf} = \left(\frac{L \uparrow_B}{\varepsilon \,\sigma}\right)^{1/4} \tag{1}$$

where  $L \uparrow_B$  is the upward infrared radiation (W m-2),  $\sigma$  is the Stefan-Boltzmann constant (W m-2 K-4), and the dimensionless emissivity  $\varepsilon$  was assumed to be 0.98 after Wilber et al. (1999). Furthermore, soil temperature (Tsoil) was measured at 2 cm soil depth in intervals of 30 minutes at an adjacent polygon rim and center.

**3.2 Soil sampling and vegetation indices**

Undisturbed A total of six soil samples were taken from the thawed\_active layer at the polygon rim using steel rings (diameter 6 cm). At the water saturated polygon center, an undisturbed soil monolith was one soil sample taken from the thawed\_active layer using a spade, and subsequently separated\_subsampled 
[revised manuscript text omitted]

The empirical Q10 model (van't Hoff, 1898) was fitted to the measured Reco and RH fluxes:

$$= R_{hase} \times O_{10} \frac{\frac{T_{a,surf,soil} - T_{ref}}{\gamma}}{\gamma}$$
(2)

where the fit parameter  $R_{base}$  was is the basal respiration at the reference temperature  $T_{ref}$  (15 °C). Tref The reference temperature and  $\gamma$  (10 °C) were held constant according to Mahecha et al. (2010).  $Q_{10}$  was a fit parameter indicating describing the ecosystem sensitivity of respiration to a 10 °C change in temperature. For this study a fixed  $Q_{10}$  value of 1.52 was used, which represents the seasonal mean value of the bulk partitioning model for the CO2 fluxes in the EC footprint area established by (Runkle et al. (2013)). Air temperature (Ta), surface temperature (Tsurf), and

- soil temperature (Tsoil) measured at a depth of 25 cm were tested as input variables.
  The model calibration was done with MATLAB® R2015a (The MathWorks Inc., Natick, MA, 2000). The model parameters were estimated by nonlinear least-squares regression fitting (nlinfit function), and the uncertainty of the parameters was determined by calculating the 95% confidence intervals using the nlparci function. The selection of the best performing temperature as input variable for the Reco and RH model was based on comparing the R2adj of the model runs with different temperatures as input variable. The selected input variable was chosen
  - for all measurement plots of the same  $\frac{\text{microsite}_{site}}{\text{might have had a better R}^2_{adj}}$ .

330

315

320

Reco,H

To estimate GPP, the chamber-measured  $R_{eco}$  fluxes were subtracted from the NEE fluxes separately-for each measurement plot. The rectangular hyperbola function was fitted to the calculated GPP fluxes as a function of PAR (in µmol m-2 s-1):

$$GPP = -\frac{P_{max} \times \alpha \times PAR}{P_{max} + \alpha \times PAR}$$
(3)

where the fit parameter  $P_{max}$  was the maximum canopy photosynthetic potential (hypothetical maximum of  $P_{max}GPP$  at infinite PAR). The values for the initial canopy quantum efficiency  $\alpha$  (dimensionlessin µg m-2 s-1 /  $\mu$ mol m-2 s-1; initial slope of the  $P_{max}$ -PAR curveGPP 
[revised manuscript text omitted]
 As such, the low net carbon CO2 uptake (NEE) at the rim occurred are caused not only because of by low GPP, but also due to by higher RecoH fluxes compared to the center. The higher NEE at the polygon center compared to the rim is mainly driven by substantially higher GPP, and lower RH fluxes, which are due to differences in second enter to the rest of the rest of the rest of the substantially higher GPP.
- 630 differences in vascular plant cover, temperature and hydrology. This finding is in good agreement with Nobrega

and Grogan (2008) who compared a wet sedge, with a dry heath, and a mesic birch site and found that the net carbon-highest CO2 uptake at the wet sedge site was highest because of due to limited respiration  $\underline{R}_{eco}$  due to associated with the water-logged conditions.

- Interestingly, measurements of CO2 fluxes at the polygon rim showed an increase of net CO2 uptake 635 NEE throughout September, whereas at the polygon center the net CO2 uptakeNEE appeared to continuously decrease (lower net uptake of CO2). This increase in late season NEE at the polygon rim cannot be explained by rising PAR or temperature, but- Rather, the increase of net CO2 uptake at the rim towards the end of September may be related to the photosynthetic activity of mosses. At the study site, Kutzbach et al. (2007b) considered-the September at the EC footprint area as period where moss photosynthesis dominates C uptakeGPP. -occurs mostly 640 due to moss photosynthesis. During this time of the growing season, mosses can still assimilate substantial amounts of  $CO_2$  because they tend to reach light saturation at lower irradiance (Harley et al., 1989). The photosynthetic activity of mosses declines rapidly when they face desiccation, because they cannot actively control their tissue water content (Turetsky et al., 2012). Additionally, Hit has been was also shown that mosses face light stress during times of high PAR (Murray et al., 1993). This light stress causes delayed senescence and more late-season 645 photosynthesis (Zona et al., 2011). Therefore, On Samoylov, the photosynthetic activity onat the moss-dominated polygon rim is expected to be low during warm and dry periods such as those seen at the beginning of September 2015, and during times of high PAR. In contrast, with continuous rainfall, dew formation and the lower PAR observed in mid-September, the mosses on the polygon rim are likely to have resumed their metabolic activity,
- 650 reported the highest contribution of mosses to GPP at the beginning and end of the growing season. With continuous rainfall, dew formation and lower PAR in mid September, the mosses are likely resume metabolic active, which led to increasing net CO2 uptake at the rim.

which led to increasing NEE at the rim. These findings are in good agreement with Olivas et al. (2011), who

**5.3 Partitioning respiration fluxes in arctic tundra ecosystems**

- To date only a handful of few studies have estimated growing season  $R_H$  fluxes from arctic tundra ecosystems over 655 a growing season under in situ conditions (Nobrega and Grogan, 2008; Biasi et al., 2014). Surprisingly, the differences in RH flux estimates reported in the literature-between these estimates and those presented in this study were rather low. Differences in  $R_{\rm H}$  fluxes measured with the trenching method may be caused result from by differences in the time between trenching and start of the measurement. Nobrega and Grogan (2008) for example started their  $R_{\rm H}$  measurements one day after clipping, while measurements in this study as well as in the study and 660 that of Biasi et al. (2014) started about one year after treatment. Therefore, even-although these studies employed a similar partitioning approach for seasonal estimates of RH fluxes was similar for all studies, any comparison must be made with caution. The few RH flux estimates reported in the literature from other arctic tundra sites were higher than the RH values from the Lena River Delta ( $0.5 \pm 0.1$  and  $0.3 \pm 0.02$  g C m-2 d-1 at polygon rim and center, respectively). Higher growing season RH fluxes than found in this study throughout the growing season  $(0.8-1.8 \text{ g C m}^{-2} \text{ d}^{-1})$  were have been measured at a mesic birch and a dry heath site at Daring Lake in Canada 665 (Nobrega and Grogan, 2008) and at a bare peat site  $(1.0 \text{ g C m}^{-2} \text{ d}^{-1})$  in the subarctic tundra at Seida, Russia (Biasi et al., 2014). Both sites contained substantially higher amounts of SOC in the organic-rich layer than the soil at the polygon rim and were well-aerated compared to the soil at the polygon center, which both mostboth of which likely caused a higher organic matter decomposition rate and could explain the higher RH fluxes than found at the
- 670 polygonal tundra micrositesites. Similar RH fluxes to those reported in our study were measured at a wet sedge site

in Daring Lake (0.4 g C m-2 d-1) (Nobrega and Grogan, 2008), where soil and environmental conditions like WT, ALD, soil temperature, vegetation and SOC were similar to the Samoylov sites and at vegetated peat sites in Seida (0.4-0.6 g C m-2 d-1) (Biasi et al., 2014). Despite these differences, the averaged contributions of  $R_H$  to  $R_{eco}$  of 42% at the center and 60% at the rim are in good agreement with those observed at Seida (37 – 64%) and Daring Lake

- (44 64%). Similar contributions were-have also been determined from an arctic tussock tundra sites, where RH makes up approximately 40% of growing season Reco (Segal and Sullivan, 2014; Nowinski et al., 2010) and from a moist acidic tussock tundra site (Hicks Pries et al., 2013). In contrast to these results, in a subarctic peatland Dorrepaal et al. (2009) -determined-report a substantially higher contribution of RH to Reco with of about 70% in a subarctic peatland. The different contributionce in the contribution of RH to Reco between at the polygon rim and
- 680 center at our study siteon Samoylov Island can be related to differences in vascular plant coverage and moisture conditions between both these micrositesites. HThe higher GPP at the center than at relative to the rim also caused also higher rates of RA-and in turn lowered-lowering the contribution of RH to Reco. Additionally, anoxic soil conditions due to standing water, like atwhich characterized the polygon center, were not favorable forreduced SOM decomposition rates of SOM. Furthermore, Moyano et al. (2013) and Nobrega and Grogan (2008) -concluded

685 have shown that consistently moderate moisture conditions, as at the polygon rim, promotes fast decomposition of SOMmicrobial activity and therefore ensures enable higher RH rates than at the center. Interestingly, aAt the polygon center, we observed significant correlations of the WT significantly correlated with

Reco and RA fluxes, but no correlation between RH fluxes and WTwas found. In contrast to this, none of the determined respiration fluxes (Reco, RH, RA) correlated with VWC at the polygon rim, which might be due to a rather low range of VWC (28 – 34 %). The RA fluxes mightmay be negatively affected by high WT due to submersion of the moss layer and partwise vascular leavesfs as submersion can lead to plant stress, reducing productivity and nutrient turnover (Gebauer et al., 1995). Low soil moisture contents can limit the growth and productivity of an ecosystem (Chen et al., 2015) as desiccation lowers the photosynthetic activity (Turetsky et al., 2012), and in turn lowers RA fluxes. (Moyano et al., 2013) However, if this RA fluxes would be reduced due to

- 695 low photosynthetic activity-were the case, we would expect a correlation between GPP and RA fluxes-WT, as observed at the polygon rim ( $R^2 = 0.48$ , p < 0.05) which was but not observed at the center ( $R^2 = 0.01$ , p > 0.05).-Instead, only half as much CO2 is released by RA at the center compared to the rim at similar GPP fluxes, as the GPP : RA ratio indicates (10.5 vs. 5.1 for the polygon center and rim, respectively). However, it is likely that RA is reduced due to the water-saturated soils as shown previously for Reco fluxes in the Arctic (e.g. Christensen et al.,
- 700 1998) maybe due to slow diffusion under water-saturated conditions (Frank et al., 1996). Furthermore, it might be possible that RH fluxes are not affected by water table fluctuations as the decomposition of SOM could take place in deeper layers. it is likely that the respired CO2 is Wetzel et al. (1984) 'recycled' due to of slow diffusion through the moss layer (Frank et al., 1996). CO2 through the water phaseEvidence for this process was already shown in polygonal ponds (Liebner et al., 2011). This finding is in contrast to a set of studies who which
- 705 attributedexplained correlations between  $R_{eco}$  fluxes and WT fluctuations solely with to the impact of oxygen availability on  $R_H$  fluxes (Juszczak et al., 2013; Chimner and Cooper, 2003; Dorrepaal et al., 2009), or observed an impact of moisture conditions on  $R_H$  fluxes across multiple peatland ecosystems (Estop-Aragonés et al., 2018), while another study has shown no effect between water table fluctuations and  $R_{eco}$  fluxes (Chivers et al., 2009). However, the partitioning approach used in this study showed that  $R_H$  fluxes are not responding to water table
- 710 fluctuations. Instead the CO2 release by  $R_A$  is correlated with water table fluctuations. However, T these findings show the importance of the soil water content hydrologic conditions for  $R_{eco}$  fluxes and the need for partitioning

approaches to understand the response of the underlying processes individual of  $R_{eco}$  fluxes on to changing hydrologic conditions.

In order tTo determine the individual impacts of hydrological conditions and temperature on the RH and RA fluxes, it would be useful to perform both warming and wetting experiments *in situ*. So far, although a number of studies have determined the temperature response of NEE, GPP, and Reco fluxes in arctic ecosystems with warming experiments (e.g. Natali et al., 2011; Frey et al., 2008; Voigt et al., 2017), however, much less research has focused on the response of RA and RH fluxes to increasinged temperatures (Hicks Pries et al., 2015). Wetting experiments in arctic tundra ecosystems to determine the individual response of RA and RH fluxes to changing hydrological conditions are also lacking so far. As climate change will likely lead to strong changes in the hydrological regimes of Siberian tundra regions (Zimov et al., 2006c; Merbold et al., 2009), the responses of respiration fluxes to altered hydrological as climate warming will likely lead to severe changes of the hydrological regimes in Siberian tundra regions (Merbold et al., 2009; Zimov et al., 2006b).

**6 Conclusion**

The contributions of GPP, Reco, RH and RA to CO2 NEE fluxes in a drained (rim) and water-saturated (center) micrositesite in the arctic polygonal tundra of northeast Siberia have been quantified in this study. Both investigated micrositesites acted as CO2 sinks during the measurement period mid-July to end of September 2015.
The polygon center acted aswas a considerably stronger CO2 sink than the polygon rim. The main drivers behind these differences in CO2 fluxes at the microsites-pedon scale were the higher GPP at the polygon center as well as lower RH and RA-fluxes at the polygon center. The substantial differences in NEE differences identified in NEE between the dry and wet tundra sites two investigated microsites highlight the importance of microscale pedon scale measurements for reliable estimates of CO2 surface-atmosphere fluxes from arctic tundra sites and the important role of soil moisture conditions on CO2 fluxes. Hereby, it was shown that RA and RH-fluxes respond differently depending on hydrological conditionswater table changes, with a low\_release of CO2 by RA fluxes during times of a high water tables. Therefore, it is recommended that future studies determining partitionedon CO2 fluxes from arctic tundra ecosystems should focus on the role of hydrological conditions as a driver of these fluxes, to obtain a more in depth insight into this relationship.

[revised manuscript text omitted]

a: this study; b: Olivas et al. (2011); cg: Oechel et al. (1995); d: Lara and Tweedie (2014); e: Lara et al. (2012); f: Nobrega and Grogan (2008); g: Kwon et al. (2016); h: Zamolodchikov et al. (2000); i: Heikkinen et al. (2004), \*: standard deviation estimated; j: Vourlitis et al. (2000); k: (Rößger et al., 2019)

1125